# Phosphorylation-mediated interactions with TOPBP1 couple 53BP1 and 9-1-1 to control the G1 DNA damage checkpoint

Nicolas Bigot[1†], Matthew Day[1†], Robert A Baldock[2‡], Felicity Z Watts[2], Antony W Oliver[1*], Laurence H Pearl[1*]

[1]Cancer Research UK DNA Repair Enzymes Group, Genome Damage and Stability Centre, School of Life Sciences, University of Sussex, Brighton, United Kingdom; [2]Genome Damage and Stability Centre, School of Life Sciences, University of Sussex, Brighton, United Kingdom

**Abstract** Coordination of the cellular response to DNA damage is organised by multi-domain 'scaffold' proteins, including 53BP1 and TOPBP1, which recognise post-translational modifications such as phosphorylation, methylation and ubiquitylation on other proteins, and are themselves carriers of such regulatory signals. Here we show that the DNA damage checkpoint regulating S-phase entry is controlled by a phosphorylation-dependent interaction of 53BP1 and TOPBP1. BRCT domains of TOPBP1 selectively bind conserved phosphorylation sites in the N-terminus of 53BP1. Mutation of these sites does not affect formation of 53BP1 or ATM foci following DNA damage, but abolishes recruitment of TOPBP1, ATR and CHK1 to 53BP1 damage foci, abrogating cell cycle arrest and permitting progression into S-phase. TOPBP1 interaction with 53BP1 is structurally complimentary to its interaction with RAD9-RAD1-HUS1, allowing these damage recognition factors to bind simultaneously to the same TOPBP1 molecule and cooperate in ATR activation in the G1 DNA damage checkpoint.
DOI: https://doi.org/10.7554/eLife.44353.001

**\*For correspondence:**
antony.oliver@sussex.ac.uk (AWO);
Laurence.Pearl@sussex.ac.uk (LHP)

†These authors contributed equally to this work

**Present address:** ‡Solent University, Southampton, United Kingdom

**Competing interests:** The authors declare that no competing interests exist.

## Introduction

53BP1 and TOPBP1 are multi-domain scaffold proteins that individually contribute to the genome stability functions of mammalian cells (*Sokka et al., 2010*; *Panier and Boulton, 2014*; *Wardlaw et al., 2014*; *Zimmermann and de Lange, 2014*). 53BP1 is recruited to sites of DNA damage through interaction with post-translationally modified histones (*Botuyan et al., 2006*; *Fradet-Turcotte et al., 2013*; *Baldock et al., 2015*; *Kleiner et al., 2015*) and is strongly implicated in the process of choice between the non-homologous end joining and homologous recombination pathways of double-strand break (DSB) repair (*Leung and Glover, 2011*; *Aparicio et al., 2014*; *Daley and Sung, 2014*), as well as the localisation of activated ATM that is required for DSB repair in heterochromatin (*Lee et al., 2010*; *Noon et al., 2010*; *Baldock et al., 2015*). Amongst other roles, TOPBP1 is involved in replication initiation through its interaction with TICCR/Treslin (*Kumagai et al., 2010*; *Boos et al., 2011*); the DNA damage checkpoint through interactions with 9-1-1, RHNO1/RHINO and ATR (*Kumagai et al., 2006*; *Delacroix et al., 2007*; *Lee et al., 2007*; *Cotta-Ramusino et al., 2011*; *Lindsey-Boltz and Sancar, 2011*); and sister chromatid de-catenation and suppression of sister chromatid exchange via interactions with TOP2A and BLM (*Blackford et al., 2015*; *Broderick et al., 2015*).

Genetic studies indicate that the fission yeast orthologues of 53BP1 and TOPBP1 – Crb2 and Rad4 (also known as Cut5) respectively – interact directly with each other, and play important roles in the DNA damage response (*Saka et al., 1997*). Subsequent studies have shown that physical

association of Crb2 and Rad4 is mediated by the interaction of a pattern of cyclin-dependent kinase phosphorylation sites on Crb2, with the BRCT domains of Rad4 (*Du et al., 2006*; *Qu et al., 2013*). A functionally important physical interaction of mammalian 53BP1 and TOPBP1 has also been observed (*Cescutti et al., 2010*), and found to depend upon the functionality of specific TOPBP1 BRCT domains, suggesting that this interaction might also mediated by phosphorylation. However, the nature and identity of the sites that might be involved were not determined.

Using a structurally-derived consensus pattern for phospho-ligand binding to the N-terminal regions of TOPBP1 and Rad4 (*Day et al., 2018*), we have now identified conserved phosphorylation sites in human 53BP1, and have biochemically and structurally characterised their interaction with BRCT domains of TOPBP1. Mutation of these sites abrogates functional interaction of 53BP1 and TOPBP1 in G1, showing them to be both necessary and sufficient for the association of the two proteins. Mutation also disrupts co-localisation of 53BP1 with the TOPBP1-associated ATR kinase, and causes defects in a G1/S checkpoint response due to disruption of downstream CHK1 signalling. The binding sites of the 53BP1 phosphorylation sites on TOPBP1 are complementary to that of the phosphorylated tail of RAD9 in the 9-1-1 checkpoint clamp. Our results delineate a key scaffold role of TOPBP1 in bringing together 53BP1 and 9-1-1 to control S-phase entry in the presence of DNA damage.

## Results

### Identification of TOPBP1-interacting phosphorylation sites in 53BP1

Previous studies demonstrated a biologically important interaction between TOPBP1 and 53BP1 – the putative metazoan homologue of Crb2 – involving BRCT1,2 and BRCT4,5 of TOPBP1 (*Cescutti et al., 2010*), although the molecular basis of this interaction was not fully determined. Although 53BP1 has many roles in mammalian cells that are distinct from those attributed to Crb2 in fission yeast, there is an emerging picture that many of the functional interactions made by these proteins are conserved. Thus, both Crb2 and 53BP1 have now been shown to interact through their C-terminal tandem $BRCT_2$ domains, with the common post-translational histone mark of DNA damage, γH2AX (*Kilkenny et al., 2008*; *Baldock et al., 2015*; *Kleiner et al., 2015*). We therefore hypothesised that phosphorylation-dependent interactions we had previously characterised between Crb2 and Rad4 might also be involved in the putative functional interaction of their mammalian homologues 53BP1 and TOPBP1.

Using a consensus motif for selective phosphopeptide-binding to BRCT domains of TOPBP1/Rad4 (*Day et al., 2018*), we searched for potential phosphorylation sites in 53BP1 that might be involved in mediating phosphorylation-dependent interactions with TOPBP1. Four matches were identified, focused on Thr334, Ser366, Ser380 and Thr670, all of which have been previously documented as phosphorylated in phospho-proteomic analyses (*Hornbeck et al., 2004*; *Sharma et al., 2014*) (*Figure 1A*). While Ser380 has previously been implicated in binding the pro-homology repair factor SCAI (*Isobe et al., 2017*), no function has been assigned to the other sites we identified. To determine whether any of these sites bound to the N-terminal segment of TOPBP1, we synthesised fluorescently labelled phospho-peptides spanning the putatively modified residues and measured their affinity for human TOPBP1 BRCT0,1,2 and BRCT4,5 constructs using fluorescence polarisation (FP) as previously described (*Rappas et al., 2011*; *Qu et al., 2013*).

Of the four peptides, only the one centred on 53BP1-pThr670 bound tightly to TOPBP1-BRCT0,1,2 ($K_d$ = 1.5 ± 0.2 μM) (*Figure 1B*). Treatment with λ-phosphatase eliminated binding, confirming that phosphorylation of the 53BP1 sequence is required (*Figure 1C*). To define which of the two BRCT domains within the TOPBP1-BRCT0,1,2 segment is primarily responsible for the interaction, we measured binding of the 53BP1-pThr670 peptide to TOPBP1-BRCT0,1,2 harbouring mutations in BRCT1 and BRCT2 that have previously been shown to eliminate binding of other specific phospho-peptides (*Boos et al., 2011*; *Rappas et al., 2011*). A high affinity interaction ($K_d$ = 1.0 ± 0.2 μM) was still observed with TOPBP1-BRCT0,1,2-K155E, in which the phospho-binding site in the BRCT1 domain is mutationally disabled, but binding was ~10 fold weaker ($K_d$ = 13.5 ± 6.2 μM) with TOPBP1-BRCT0,1,2-K250E, in which BRCT2 is mutated. These data implicate BRCT2 as the primary binding site, consistent with the presence of a hydrophilic residue −4 to the

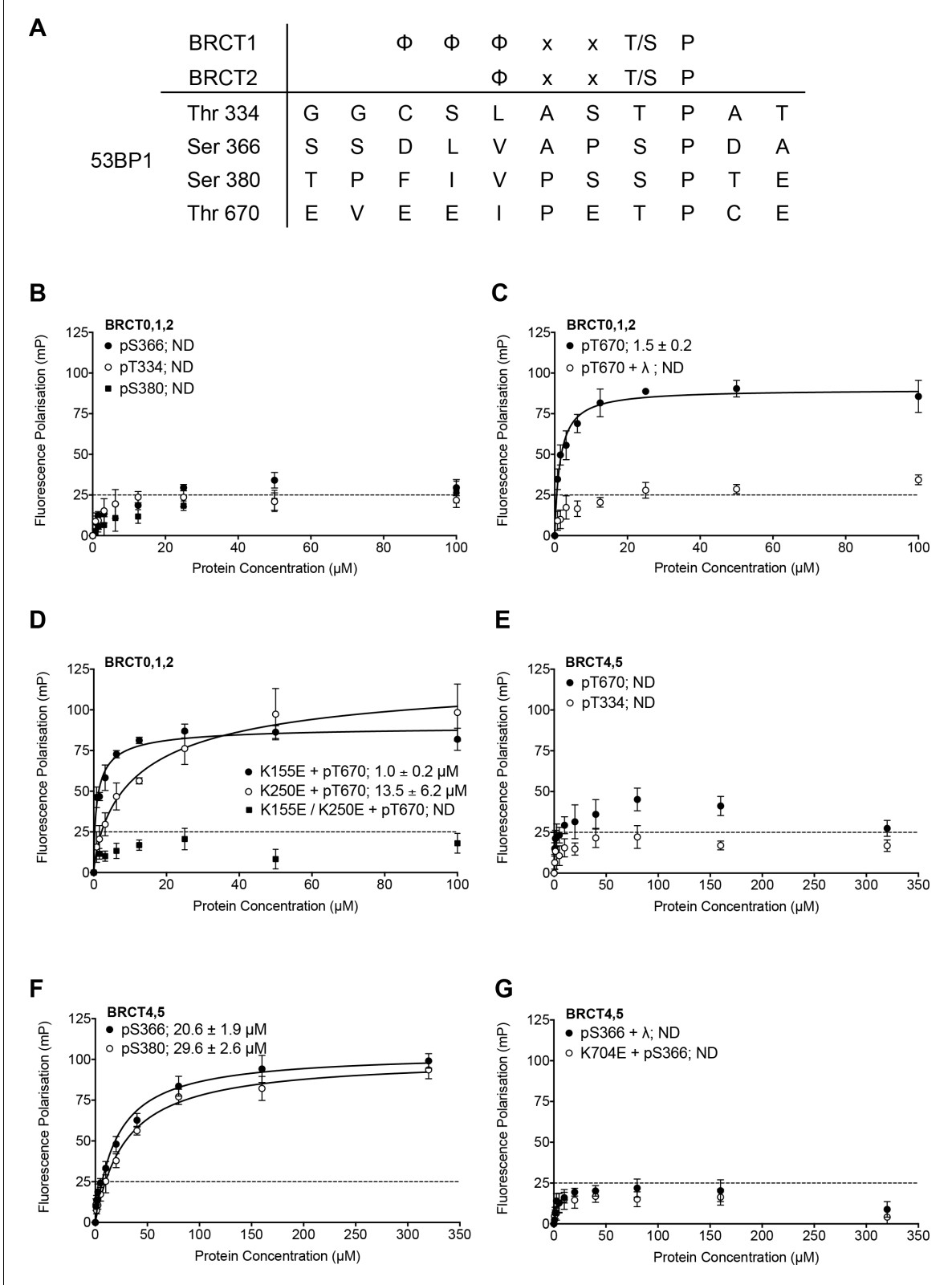

**Figure 1.** Identification and in vitro characterisation of TOPBP1 binding phosphorylation sites in 53BP1. (**A**) The TOPBP1/Rad4-binding motif matches only four potential sites out of over two hundred phosphorylation sites documented for 53BP1 (*Hornbeck et al., 2004*; *Sharma et al., 2014*). (**B**) Fluorescence polarisation (FP) analysis shows no substantial interaction of fluorescently-labelled phospho-peptides derived from the putative phosphorylation sites centred on 53BP1-Thr344, Ser366 or Ser380 with the BRCT0,1,2 segment of TOPBP1. (**C**) A fluorescently-labelled phospho-

*Figure 1 continued on next page*

Figure 1 continued

peptide centred on 53BP1-Thr670 binds to the BRCT0,1,2 segment of TOPBP1 with high affinity in FP assays. Treatment with λ-phosphatase abolishes binding, confirming that the interaction is specific for the phosphorylated peptide. (D) Charge-reversal mutation of Lys155, which is implicated in phospho-binding in BRCT1 has little effect on binding of the pThr670 peptide to BRCT0,1,2, whereas mutation of the equivalent residue, Lys250 in BRCT2 substantially decreases the affinity. Mutation of both sites completely abolishes the interaction. (E) No binding of 53BP1-derived phospho-peptides centred on Thr670 or Thr334 was detected with the BRCT4,5 segment of TOPBP1. (F) Fluorescently-labelled phospho-peptides centred on 53BP1-Ser366 and Ser380 bind with modest affinity to the TOPBP1 BRCT4,5 segment. (G) Treatment of the 53BP1-Ser366 phosphopeptide with λ-phosphatase or charge-reversal mutation of Lys704, which is implicated in phospho-binding in BRCT5, abolishes interaction of the phosphopeptide to BRCT4,5.

DOI: https://doi.org/10.7554/eLife.44353.002

phosphothreonine (*Day et al., 2018*), but with BRCT1 potentially providing a weaker alternative site. Mutation of both BRCT phospho-binding sites eliminated interaction (*Figure 1D*).

Given the previous implication of a role for BRCT4,5 in mediating the interaction of TOPBP1 and 53BP1 (*Cescutti et al., 2010*), we also looked at the ability of this segment of TOPBP1 to bind these 53BP1-derived phosphopeptides. In contrast to its tight interaction with TOPBP1-BRCT0,1,2, the 53BP1-pThr670 peptide showed little binding to TOPBP1-BRCT4,5, nor did the 53BP1-pThr334 peptide (*Figure 1E*). However, the 53BP1-pSer366 peptide did show a clear interaction with TOPBP1-BRCT4,5, albeit with a more modest affinity ($K_d$ = 20.6 ± 1.9 μM). 53BP1-pSer380 also bound TOPBP1-BRCT4,5 but with still lower affinity ($K_d$ = 29.6 ± 2.6 μM) (*Figure 1F*). Treatment of the phosphopeptide with λ-phosphatase or mutation of Lys704, which is implicated in phosphate recognition in TOPBP1-BRCT4,5 eliminated binding, confirming the phospho-specificity of the interaction (*Figure 1G*).

## Structure of TOPBP1 – 53BP1 phospho-peptide complexes

To further characterise the nature of these interactions we sought to obtain crystal structures of the TOPBP1-53BP1 phospho-peptide complexes. Co-crystallisation of TOPBP1-BRCT0,1,2 with the 53BP1-pThr670 phospho-peptide produced a crystal structure with peptide bound to BRCT2, consistent with the specificity it displays for that site in binding assays (*Figure 2A*). The position and overall conformation of the bound peptide is similar to that previously described for the interaction of Crb2-derived phospho-peptides pThr187 and pThr235 bound to BRCT2 of Rad4 (*Qu et al., 2013*). The phosphorylated side chain of 53BP1-Thr670 interacts with TOPBP1-Thr208 and TOPBP1-Lys250, which are topologically conserved in all BRCT domains capable of phospho-specific peptide interaction (*Leung and Glover, 2011*). The phosphate also interacts with the head group of TOPBP1-Arg215 which makes an additional hydrogen bond interaction with the side chain of 53BP1-Glu669. The side chain of the −3 residue of the phospho-peptide motif (see above), 53BP1-Ile667, packs into a hydrophobic pocket formed between the side chains of TOPBP1 residues Leu233, Gln249, Lys250 and Cys253. Additional specificity is provided by polar interactions between 53BP1-Glu669 and TOPBP1-Arg215, 53BP1-Glu666 and TOPBP1-Lys234, and 53BP1-Glu665 and the side chains of Arg256 and Trp257 of TOPBP1 (*Figure 2B*).

We were also able to obtain co-crystals of the 53BP1-pSer366 peptide bound to TOPBP1-BRCT4,5 (*Figure 2C*). The 53BP1-pSer366 peptide binds to BRCT5 in a similar position and orientation as previously observed for phospho-peptide binding to TOPBP1-BRCT1 and BRCT2 (see above) and Rad4-BRCT1 and BRCT2 (*Qu et al., 2013*) (*Figure 2D*). The phosphoserine side chain engages with the topologically conserved TOPBP1-Ser654 and with the side chain of TOPBP1-Lys704. The −3 residue in the bound peptide, 53BP1-Val363, binds into a hydrophobic pocket lined by the side chains of TOPBP1 residues Phe673, Ser703, Lys704, Ala707 and Trp711. Additional specificity is provided by the hydrophobic packing of the side chains of 53BP1 residues Ala364 and Pro365 around the protruding side chain of TOPBP1-Tyr678, and the packing of 53BP1-Leu362 into a hydrophobic pocket formed by the side chains of TOPBP1-Tyr678, Val680 and Met689.

This mode of binding, conserved in Rad4 and N-terminal TOPBP1 BRCT domain interactions, is markedly different from that reported for binding of an MDC1-derived phospho-peptide to TOPBP1-BRCT4,5 (*Leung et al., 2013*). In that study, the peptide runs perpendicular to the orientation of the 53BP1-peptide observed here, and the phosphorylated residue of the ligand peptide does not appear to interact with the canonical phospho-interacting residues of TOPBP1-BRCT5.

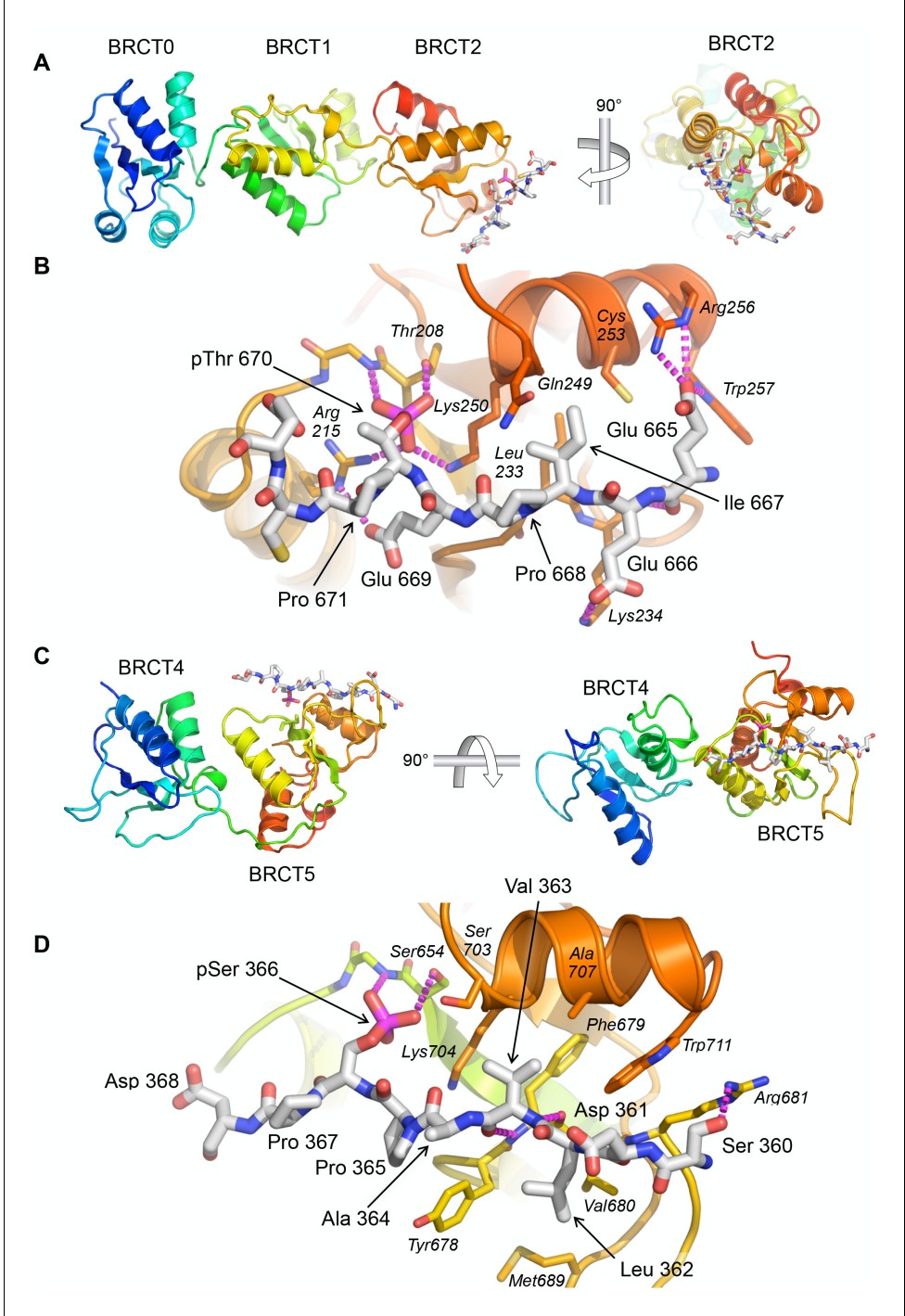

**Figure 2.** Crystal structures of TOPBP1 – 53BP1 phosphopeptide complexes. (**A**) Structure of TOPBP1 BRCT0,1,2 bound to a 53BP1-pT670 peptide. As predicted from the consensus motif and confirmed by the FP data, this peptide binds to BRCT2. TOPBP1 secondary structure is rainbow-coloured (N-terminus blue - > C terminus red). (**B**) Interactions of 53BP1-pT670 peptide and TOPBP1-BRCT2. Dashed lines indicate hydrogen bonding interactions. See text for details. (**C**) Structure of TOPBP1 BRCT4,5 bound to a 53BP1-pS366 peptide. Consistent with the FP data, the peptide binds to BRCT5. (**D**) Interactions of 53BP1-pS366 peptide and TOPBP1-BRCT5. Dashed lines indicate hydrogen bonding interactions. See text for details.

DOI: https://doi.org/10.7554/eLife.44353.003

Interestingly, TOPBP1-BRCT4,5 has recently been identified as the binding site for a phosphorylated motif in the RECQ-family helicase BLM (*Blackford et al., 2015*) and a crystal structure of the complex determined (*Sun et al., 2017*). The identified binding site in BLM - 300-FVPPpSPE-306 shows strong similarity to the 53BP1-motif we have identified - 362-LVAPpSPD-368, and binds to BRCT5 in a very similar manner, but quite differently from the proposed binding mode of the MDC1 SDT motif (*Leung et al., 2013*), which we have recently shown plays no role in mediating MDC1 interaction with TOPBP1 (*Leimbacher et al., 2019*). The similarity of the sequences surrounding BLM-pSer304 and 53BP1-pSer366, especially the −1 proline, the small hydrophobic residues at −2 and −3, and the large hydrophobic residue at −4, suggests a binding specificity for BRCT5 that is related to, but subtely distinct from those defined for BRCT1 and BRCT2; further examples will be required to confirm this.

## TOPBP1-binding sites on 53BP1 are phosphorylated in vivo

To verify the presence of these TOPBP1-binding phosphorylation sites in cells, we generated independent phospho-specific antibodies to the two sites and looked at the presence of immuno-reacting species in HeLa cell lysates, both before and after DNA damage by ionising radiation (IR). For both $\alpha$−53BP1-pSer366 and $\alpha$−53BP1-pThr670 antibodies, clear reactive bands were seen following IR, which were weaker in non-irradiated cells, and considerably diminished by siRNA knockdown of 53BP1 (*Figure 3A*). We also performed immunofluorescence with these antibodies in irradiated U2OS cells (*Figure 3B*) and observed strong co-localisation of foci formed by the phospho-specific antibodies with those formed by antibodies to total 53BP1, consistent with the endogenous protein being phosphorylated on Ser366 and Thr670. To confirm the specificity of the antibodies for the phosphorylated state of their target sites, we knocked-down endogenous 53BP1 in siRNA-resistant wild-type eYFP-53BP1 U2OS cells harbouring either the WT 53BP1 or the S366A or T670A 53BP1 mutants. While we saw strong coincidence of the eYFP signal with immunofluorescent foci to total 53BP1 in the wild-type constructs, this was lost for both of the phospho-specific antibodies in cells transfected with 53BP1 mutated in their respective phosphorylation site epitope (*Figure 3C*).

## pSer366 and pThr670 mediate 53BP11 interactions with TOPBP1 in vivo

To determine whether the interactions of 53BP1-pSer366 and 53BP1-pThr670 with the BRCT domains of TOPBP1 that we characterised in vitro, are important for functional interactions of TOPBP and 53BP1 in vivo, we knocked-down 53BP1 by siRNA in U2OS and RPE1 cells (*Figure 3—figure supplement 1*), and transfected them with vectors expressing siRNA-resistant eYFP-53BP1 constructs. In both cell types expressing tagged wild-type eYFP-53BP1 constructs, TOPBP1 and eYFP-53BP1 co-localised into damage foci in G1 (Cyclin A negative) cells following irradiation, reflecting the direct interactions we observed in vitro. However, while irradiated U2OS cells expressing the 53BP1-S366A and/or T670A mutants, still formed eYFP-53BP1 foci, formation of TOPBP1 foci coincident with these was significantly reduced (*Figure 4A,B*). A very similar effect was seen in RPE1 cells (*Figure 4—figure supplement 1A*). These data suggest that both of these phosphorylation sites are required for maintaining the interaction between 53BP1 and TOPBP1 and for recruiting TOPBP1 into 53BP1 damage foci in G1 cells negative for cyclin A.

## pSer366 and pThr670 facilitate activation of the ATR checkpoint kinase cascade

As interaction of 53BP1 and TOPBP1 has previously been suggested to play a role in G1 checkpoint function, we established a dual EdU/BrdU labelling checkpoint assay (*Cescutti et al., 2010*) in U2OS and RPE1 cells, which allows measurement of DNA synthesis in cells moving from G1 into S-phase in the presence/absence of DNA damage. We then used this to test the role, if any, of the 53BP1 phosphorylation sites we have identified, in the G1 checkpoint response. Wild-type U2OS and RPE1 cells both showed a G1/S checkpoint in response to ionising radiation (IR), with few cells moving into DNA synthesis (visualised by BrdU incorporation) as compared to untreated cells. However, this G1/S checkpoint was greatly reduced in cells of either type in which 53BP1 protein levels were knocked down by siRNA, confirming a critical role for 53BP1 (*Figure 4C*, *Figure 4—figure supplement 1B,C*, *Figure 4—figure supplement 2A*). The checkpoint response to IR in both cell types was largely

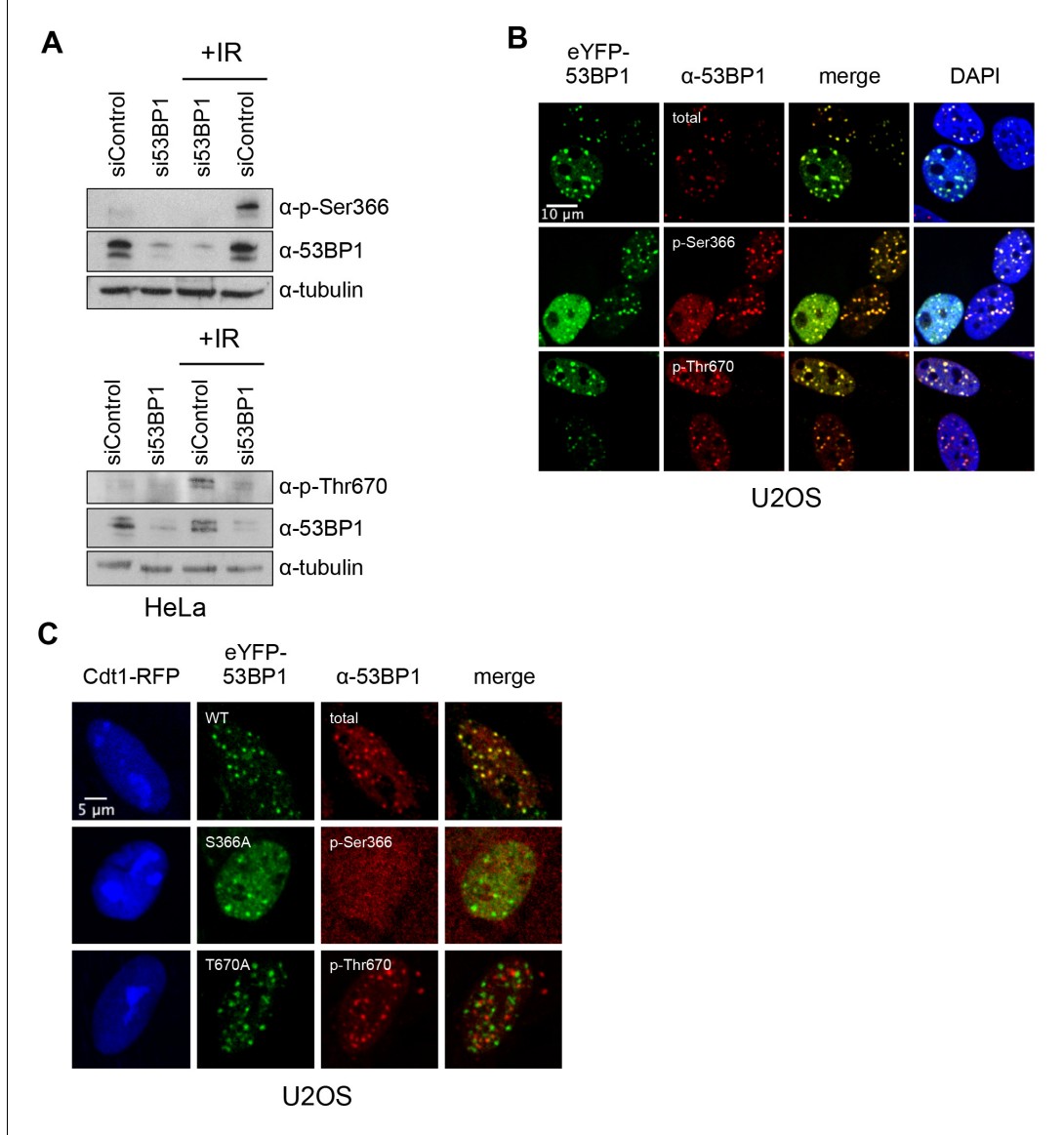

**Figure 3.** TOPBP1-binding sites on 53BP1 are phosphorylated in vivo. (**A**) Western blot of cell lysate from HeLa cells, showing induction of phosphorylation of 53BP1-Ser366 (top) and 53BP1-Thr670 (bottom) following irradiation. siRNA knockdown of 53BP1 eliminates the reactive bands in both cases, confirming the specificity of the antibody for 53BP1. (**B**) Imaging of irradiated eYFP-53BP1 WT U2OS cells with siRNA knockdown of endogenous 53BP1. 53BP1-pSer366 and 53BP1-pThr670 immunofluorescence signals co-localise in discrete foci with eYFP-53BP1 after IR (9Gy). Scale bar, 10 μm. (**C**) 53BP1-pSer366 and 53BP1-pThr670 immunofluorescent foci coincident with eYFP-53BP1 WT are lost in irradiated 53BP1 siRNA knocked-down stable eYFP-53BP1 U2OS cells expressing the S366A and T670A mutants, respectively. The α−53BP1-pThr670 antiserum has some additional low-affinity off-target reactivity unrelated to 53BP1 which is not evident when 53BP1 is present. The CDT1-RFP signal in nuclei indicates cells in G1. Scale bar, 5 μm.

DOI: https://doi.org/10.7554/eLife.44353.004

The following figure supplement is available for figure 3:

**Figure supplement 1.** Validation of 53BP1 siRNA western blots demonstrating depletion of 53BP1 protein in U2OS and RPE1 cells treated with siRNA targeted to 53BP1.

DOI: https://doi.org/10.7554/eLife.44353.005

restored in the 53BP1 knockdown cells by expression of an siRNA-resistant 53BP1 construct, but not by 53BP1 constructs with S366A and/or T670A mutations (*Figure 4D*, *Figure 4—figure supplement 1D,E*, *Figure 4—figure supplement 2B*). Consistent with this, we found that the cell cycle distribution of irradiated U2OS cells expressing the mutants were significantly perturbed compared with

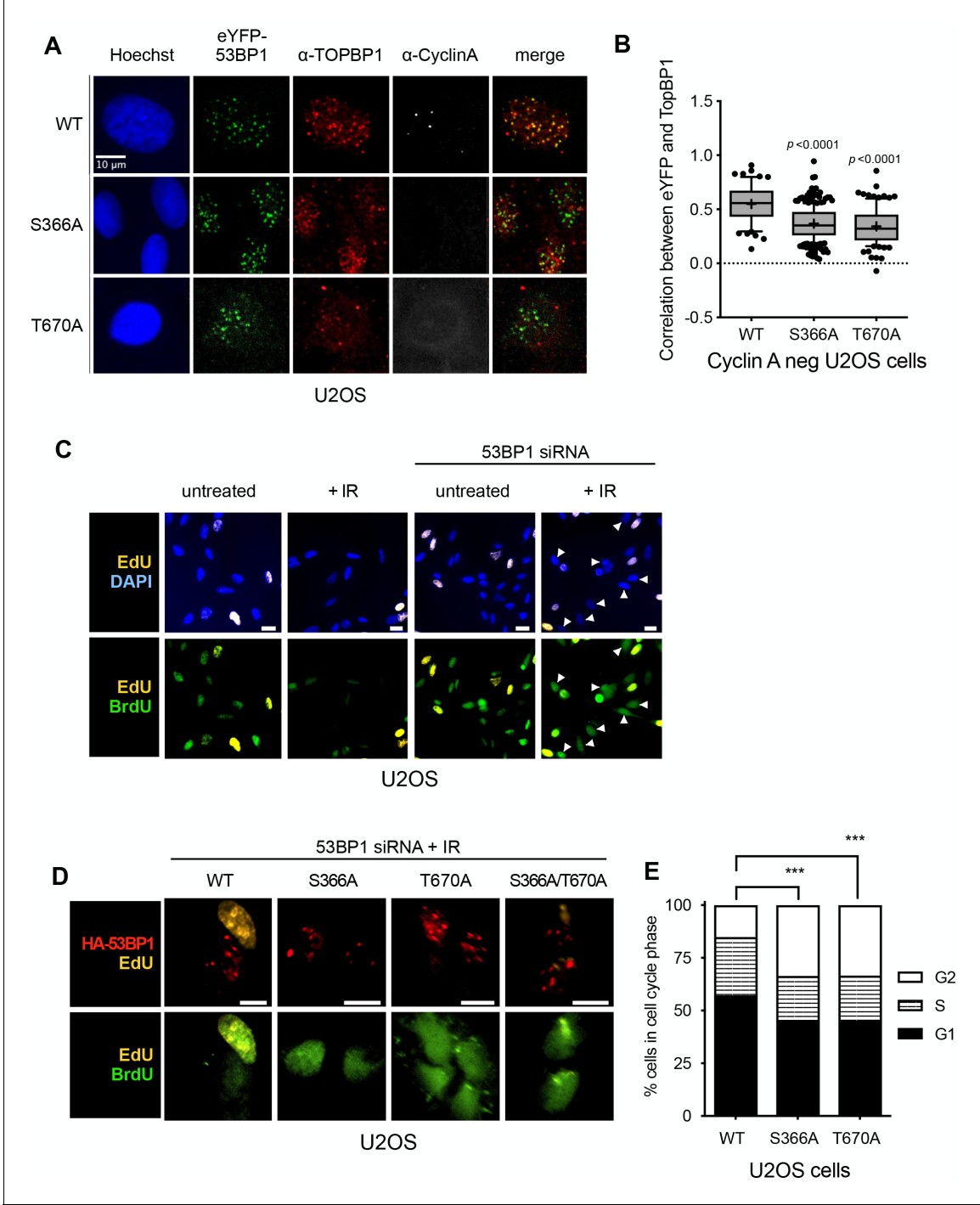

**Figure 4.** 53BP1 phosphorylation sites mediate interaction with TOPBP1 in vivo. (**A**) Four hours after 9Gy IR, TOPBP1 foci co-localise with eYFP-53BP1 WT in stably transfected U2OS cells depleted for endogenous 53BP1. Formation of co-localising TOPBP1 foci is greatly reduced in cells expressing eYFP-53BP1 S366A and T670A mutations, and the general distribution of TOPBP1 is more diffuse. The absence of substantial cyclin A immunofluorescence marks the nuclei of cells in G1. Scale bar, 10 μm. Comparable data for RPE1 cells is shown in *Figure 4—figure supplement 1*. (**B**) Statistical analysis of TOPBP1 and eYFP-53BP1 foci co-localisation per nucleus in irradiated G1 U2OS cells exemplified in **A**). Cells expressing S366A or T670A mutant eYFP-53BP1 show significantly lower levels of coincidence between TOPBP1 and eYFP-53BP1. More than 200 nuclei were counted per case. Median, mean (+), 10–90 percentiles and outliers are represented in boxplots. *p* values for the mutants relative to wild-type were calculated by a Kruskal-Wallis test corrected by Dunn's multiple comparison test. (**C**) Effect of siRNA depletion of 53BP1 on S phase entry by incorporation of BrdU (green) following damage in U2OS cells. Cells that were already in S-phase prior to DNA damage incorporate EdU (yellow) and are not further analysed. Wild-type G1 cells (EdU-) show a robust G1/S checkpoint following irradiation, do not progress into S-phase and do not incorporate BrdU. G1

*Figure 4 continued on next page*

*Figure 4 continued*

cells (EdU-) in which 53BP1 is knocked-down fail to checkpoint and progress into S-phase BrdU. EdU-/BrdU+ cells are indicated with arrowheads. Scale bars indicate 10 μm. Comparable data for RPE1 cells is shown in *Figure 4—figure supplement 1*. (D) 53BP1 siRNA knocked-down cells transfected with wild-type siRNA resistant HA-53BP1 show a G1/S checkpoint following irradiation, while those transfected with 53BP1 in which one or both TOPBP1-binding phosphorylation sites Ser 366 and Thr 670 are mutated, fail to checkpoint and progress into S-phase, incorporating BrdU. Cells that were in S-phase prior to irradiation incorporate EdU (yellow) and are not further analysed. Scale bars indicate 10 μm. Comparable data for RPE1 cells is shown in *Figure 4—figure supplement 1*. (E) Histogram of U2OS cells depleted of endogenous 53BP1 by siRNA, and transfected with either wild-type HA-53BP1 (WT) or HA-53BP1 with phosphorylation site mutants. The cell cycle phase distributions in the cells expressing mutant 53BP1 are significantly different (Chi-squared test) from that of the wild-type, with a shorter S-phase, and more cells in G2, consistent with a defective G1/S DNA damage checkpoint allowing progression into DNA replication in the presence of unrepaired damage.
DOI: https://doi.org/10.7554/eLife.44353.006

The following figure supplements are available for figure 4:

**Figure supplement 1.** TOPBP1-53BP1 co-localization and G1/S Checkpoint defects in RPE1 cells.
DOI: https://doi.org/10.7554/eLife.44353.007
**Figure supplement 2.** TOPBP1-53BP1 co-localization and G1/S Checkpoint defects in U2OS cells.
DOI: https://doi.org/10.7554/eLife.44353.008

wild-type, with the mutants displaying fewer cells in G1 or S phases and a higher proportion in G2 (*Figure 4E*).

Conditional cell-cycle progression into S-phase is controlled by a G1/S checkpoint phosphorylation cascade coupled to p53 activation and p21 up-regulation, triggered by activation of CHK2 and/ or CHK1 downstream of the primary DNA damage-sensing kinases ATM and/or ATR. As multiple components of this network are known to interact with 53BP1 and/or TOPBP1, that disruption of the phosphorylation-mediated 53BP1-TOPBP1 interaction permits progression into S-phase in the presence of DNA damage, suggests that it plays an important role in the activation of this system.

ATM and ATR have both been previously shown to form nuclear foci in G1 cells following irradiation (for example *Adams et al., 2006*; *Noon et al., 2010*; *Gamper et al., 2013*; *Averbeck et al., 2014*; *Baldock et al., 2015*). Using validated phospho-specific antibodies to the activated states of the proteins (*Noon et al., 2010* and *Figure 5—figure supplements 1* and *2*) we found that both pATM and pATR display distinct immunofluorescent foci that co-localise with 53BP1 and TOPBP1 in irradiated G1 U2OS cells expressing wild-type eYFP-53BP1. Unlike 53BP1 knockdown or point mutations in the C-terminal tandem BRCT domain of 53BP1 that affect its interaction with γH2AX (*Baldock et al., 2015*) mutation of Ser366 and/or Thr670 had no significant effect on pATM focus formation and co-localisation with 53BP1 in U2OS or RPE1 cells in G1 (*Figure 5—figure supplement 1*). This suggests that TOPBP1, which no longer co-localises with 53BP1 bearing S366A and/or T670A mutations, plays no significant role in ATM activation or localisation.

In marked contrast, co-localisation of pATR foci with 53BP1 was significantly disrupted (*Figure 5A,B*) by the S366A and/or T670A mutations of 53BP1. CHK1, which is activated by ATR in the DNA damage checkpoint response, has also been previously shown to form nuclear foci following irradiation (*Peddibhotla et al., 2011*; *Burdak-Rothkamm et al., 2015*; *Antonczak et al., 2016*). Using antibodies to total CHK1, we also observed distinct CHK1 foci co-incident with wild-type eYFP-53BP1 in 53BP1-knockdown G1 RPE1 cells following irradiation, but not in irradiated cells in which 53BP1 was knocked down by siRNA, where CHK1 displayed a diffuse pan-nuclear distribution (*Figure 5C*). CHK1 foci were restored in 53BP1 knockdown cells expressing siRNA-resistant wild-type eYFP-53BP1, but not in cells expressing eYFP-53BP1 with S366A and/or T670A mutations, which displayed the same diffuse distribution of CHK1 as the 53BP1 knockdown cells (*Figure 5D*). While CHK1, unlike ATR, is not a direct ligand of TOPBP1, these data suggest that its recruitment to sites of DNA damage in G1 and its activating interaction with ATR, most likely mediated by claspin (*Liu et al., 2006*), is dependent on the 53BP1-TOPBP1 interaction. Consistent with this functional disruption of ATR and CHK1, we observed significant decreases in DNA damage induced TP53-Ser15 phosphorylation (*Shieh et al., 2000*; *Helt et al., 2005*) and total TP53 and p21 levels, in 53BP1 knockdown G1 U2OS cells expressing S366A or T670A eYFP-53BP1, compared to wild-type (*Figure 5E,F,G,H*).

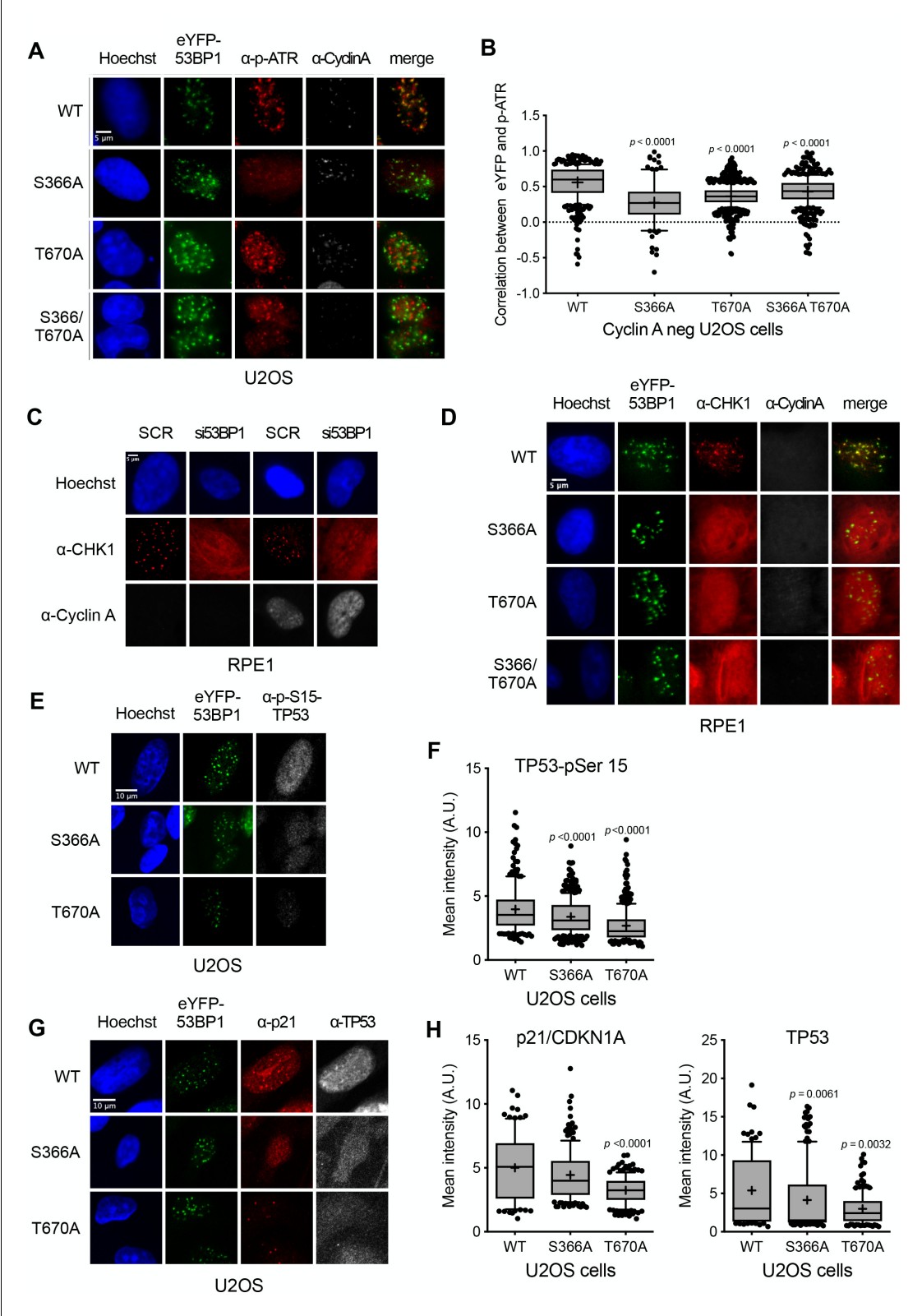

**Figure 5.** Damage checkpoint signalling through ATR is perturbed in 53BP1 phosphorylation site mutants. (**A**) ATR activated by phosphorylation on Thr1989 (pATR) forms immunofluorescent foci that co-localise with transfected eYFP-53BP1 in irradiated G1 U2OS cells with siRNA knockdown of endogenous 53BP1. However, co-localisation of ATR foci with eYFP-53BP1 foci is lost in cells expressing eYFP-53BP1 constructs with S366A and T670A mutations. The absence of substantial cyclin A immunofluorescence marks the nuclei of cells in G1. Scale bar, 5 μm. (**B**) Statistical analysis of pATR and

*Figure 5 continued on next page*

Figure 5 continued

eYFP-53BP1 foci co-localisation per nucleus in irradiated G1 U2OS cells exemplified in (**A**) More than 100 nuclei were counted per case. Median, mean (+), 10–90 percentiles and outliers are represented in boxplots. *p* values for the mutants relative to wild-type were calculated by a Kruskal-Wallis test corrected by Dunn's multiple comparison test. (**C**) CHK1 forms distinct immunofluorescent foci in irradiated G1 RPE1 cells transfected with a control scrambled siRNA (SCR). On siRNA knockdown of 53BP1, CHK1 no longer forms discrete foci, but takes on a diffuse pan nuclear distribution. The absence of substantial cyclin A immunofluorescence marks the nuclei of cells in G1. Scale bar, 5 μm. (**D**) CHK1 focus formation in irradiated G1 RPE1 cells with siRNA knockdown of endogenous 53BP1, is rescued by expression of siRNA-resistant wild-type eYFP-53BP1 but not by eYFP-53BP1 constructs with S366A and T670A mutations. Scale bar, 5 μm. (**E**) Phosphorylation of the ATR/CHK1 target site, TP53-Ser15, is evident in the nuclei of irradiated U2OS cells stably expressing the wild-type eYFP-53BP1 and depleted for endogenous 53BP1. This signal is significantly diminished in cells expressing eYFP-53BP1 constructs with S366A or T670A mutations. Scale bar, 10 μm. (**F**) Statistical analysis of mean α-TP53-Ser15 immunfluorescence per nucleus in irradiated G1 U2OS cells exemplified in (**E**) More than 100 nuclei were counted per case. Median, mean (+), 10–90 percentiles and outliers are represented in boxplots. *p* values for the mutants relative to wild-type were calculated by a Kruskal-Wallis test corrected by Dunn's multiple comparison test. (**G**) p21/CDKN1A and TP53 nuclear signals are decreased after irradiation in U2OS cells expressing eYFP-53BP1 S366A and T670A mutants and depleted of endogenous 53BP1 compared to a wild-type eYFP-53BP1 control. Neither the TP53-pSer15 (**E**) nor total TP53 immunofluorescence signals show any pattern of co-localisation with 53BP1, confirming that direct interaction of the two proteins is not significant in the context of DNA damage signalling (*Cuella-Martin et al., 2016*). Scale bar, 10 μm. (**H**) Statistical analyses of mean α-p21/CDKN1A (left) and α-TP53 (right) immunfluorescence per nucleus in irradiated G1 U2OS cells exemplified in **G**) More than 100 nuclei were counted per case. Median, mean (+), 10–90 percentiles and outliers are represented in boxplots. *p* values for the mutants relative to wild-type were calculated by a Kruskal-Wallis test corrected by Dunn's multiple comparison test.

DOI: https://doi.org/10.7554/eLife.44353.009

The following figure supplements are available for figure 5:

**Figure supplement 1.** pATM focus colocalisation with 53BP1 is unaffected by mutations in Ser366 or Thr670.

DOI: https://doi.org/10.7554/eLife.44353.010

**Figure supplement 2.** Validation of pATR antibody in U2OS cells.

DOI: https://doi.org/10.7554/eLife.44353.011

## 53BP1 and 9-1-1 co-localise in G1 damage foci

ATR localisation to DNA damage sites in S and G2 phases, requires interaction of its constitutive partner, ATRIP, with RPA-bound segments of ssDNA, arising through replication fork collapse, or resection of DSBs by MRN as a prelude to repair by HR (*Maréchal and Zou, 2015*). Activation of ATR additionally requires interaction with the AAD-domain of TOPBP1, which is recruited to damage sites via interaction with the phosphorylated C-terminus of RAD9 (*Lee et al., 2007*), within the toroidal RAD9-RAD1-HUS1 (9-1-1) checkpoint clamp (*Doré et al., 2009*), itself loaded at the inner margin of the ssDNA segment by the RAD17-RFC clamp loader complex (*Ohashi and Tsurimoto, 2017*). As our biochemical and structural analysis shows that the critical phosphorylation sites on 53BP1 selectively interact with BRCT domains 2 and 5 of TOPBP1, binding of 53BP1 to a TOPBP1 molecule would not in principle preclude simultaneous binding of the RAD9-tail, which is strongly selective for BRCT1 (*Rappas et al., 2011*; *Day et al., 2018*). We therefore considered the possibility that TOPBP1 might be able to interact simultaneously with 53BP1 and RAD9 (*Figure 6A*).

To explore this, we looked at the distributions of 53BP1 and 9-1-1 (via RAD9) in the nuclei of irradiated G1 RPE1 cells, and observed substantial coincidence of 53BP1 and RAD9 foci (*Figure 6B*). To determine whether this co-localisation reflects 53BP1 and 9-1-1 being in physical proximity within a site of DNA damage, we utilised a cellular proximity ligation assay (PLA) (Duolink, Sigma-Aldrich UK – see Materials and methods), which generates a fluorescent focus when two target proteins are within 30–40 nm of each other. We observed a strong PLA signal between RAD9 and 53BP1 in irradiated cells (*Figure 6C*), which increased significantly on exposure to IR (*Figure 6D,E*), with the major effect occurring in cells featuring the lowest Hoechst intensity typical of G1 cells with low DNA content. When TOPBP1 levels were knocked down by siRNA, we observed a significant loss of the proximity signal between RAD9 and 53BP1 in irradiated cells (*Figure 6F*), consistent with TOPBP1 playing a pivotal role in bringing these two damage recognition proteins into proximity. Similar results were obtained with U2OS cells (*Figure 6—figure supplement 1*). We were also able to detect PLA signals between RAD9 and eYFP in U2OS cells with 53BP1 siRNA knockdown in which eYFP-53BP1 was transiently transfected. However, consistent with the role of pSer366 and pThr670 in mediating recruitment of the eYFP-53BP1 to TOPBP1 and thereby bringing it into close proximity to RAD9, the RAD9-eYFP PLA signal was significantly diminished when the cells were transfected with eYFP-53BP1 constructs with S366A and T670A mutations (*Figure 6G*).

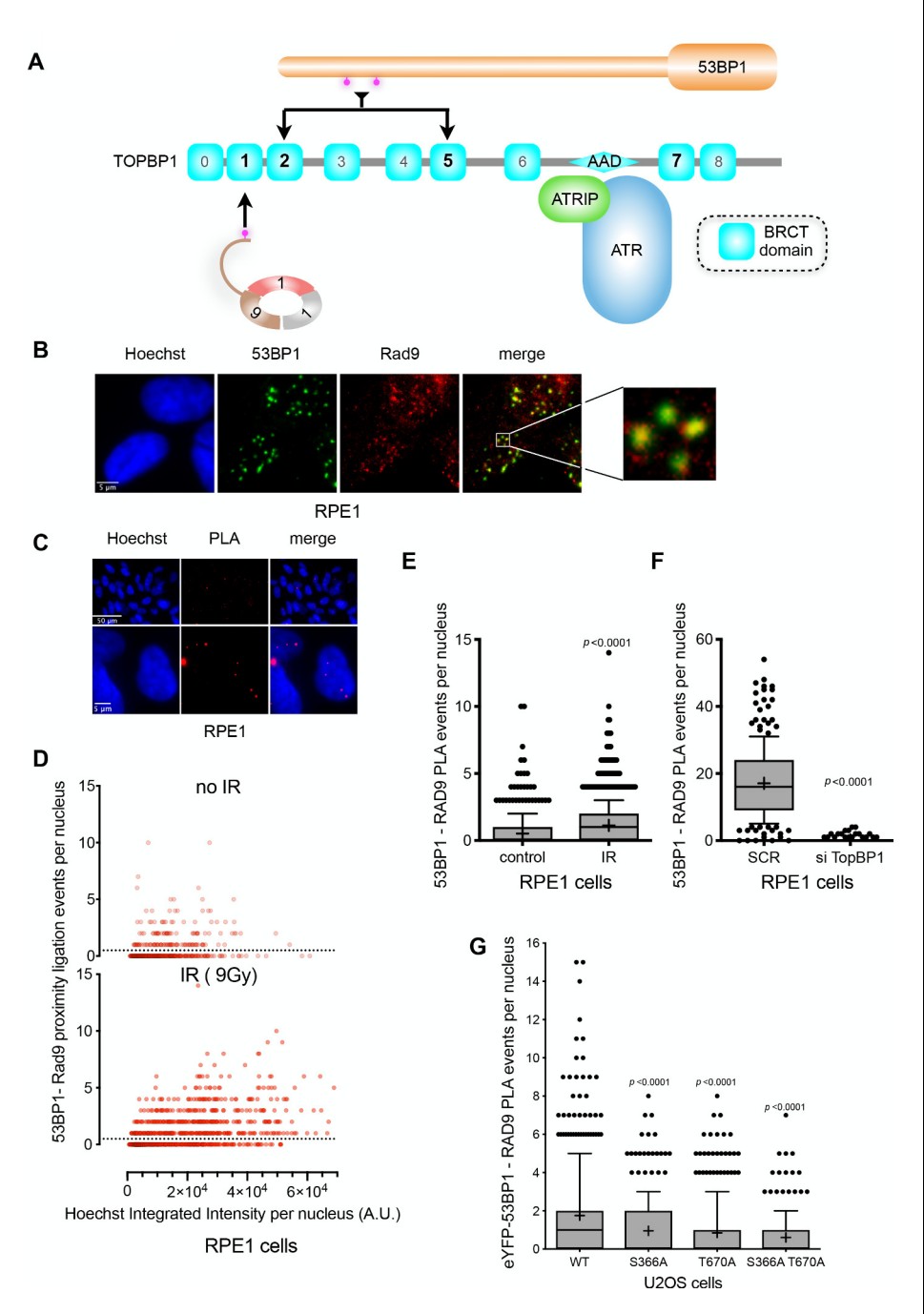

**Figure 6.** TOPBP1 physically couples 9-1-1 and 53BP1 complexes. (**A**) Schematic of domain architecture of TOPBP1 and interactions. The selective phosphorylation-dependent interactions of 9-1-1 and 53BP1 with different BRCT domains allow for the possibility of their simultaneous interaction with a single TOPBP1 molecule and their collaborative participation in ATR interaction. (**B**) 53BP1 and RAD9 immunofluorescence foci partially co-localise in irradiated RPE1 cells. Scale bar, 5 µm. (**C**) Proximity ligation assay (PLA) events (red) for RAD9 and 53BP1 demonstrating the occurrence of RAD9 and 53BP1 molecules within 30–40 nM of each other within the nuclei of irradiated RPE1 cells. Scale bar representing 50 µm and 5 µm are indicated. (**D**) Scatter plot of PLA events per nucleus for RAD9 – 53BP1 proximity as a function of nuclear Hoechst signal for irradiated (top) and non-irradiated (bottom) RPE1 cells. The PLA signal is predominantly seen in G1 cells (lower Hoechst staining) and is significantly increased by irradiation of the cells. (**E**) Statistical analysis of PLA events per nucleus in irradiated RPE1 cells shown in **D**,) showing a significant increase in PLA signals on irradiation. More than 500 nuclei were counted per case. Median, mean (+), 10–90 percentiles and outliers are represented in boxplots. $p$ values for the irradiated versus non-irradiated cells were calculated by a Mann-Whitney test. (**F**) Statistical analysis of PLA events per nucleus in irradiated RPE1 cells transfected either with a control scrambled siRNA (SCR) or an siRNA directed against TOPBP1 (*Figure 6—figure supplement 1C*). A very significant decrease in PLA signal between 53BP1 and RAD9 when TOPBP1 is knocked down. More than 200 nuclei were counted per case. Median, mean (+), 10–90 percentiles

*Figure 6 continued on next page*

Figure 6 continued

and outliers are represented in boxplots. p values for the irradiated versus non-irradiated cells were calculated by a Mann-Whitney test. Comparable data for U2OS cells is presented in *Figure 6—figure supplement 1B*. (G) Statistical analysis of PLA events per nucleus between RAD9 and eYFP, in irradiated U2OS cells with siRNA knockdown of endogenous 53BP1, transfected with either wild-type eYFP-53BP1 or eYFP-53BP1 with S366A and/or T670A mutations. More than 200 nuclei were counted per case. Median, mean (+), 10–90 percentiles and outliers are represented in boxplots. p values for the mutant versus wild-type eYFP-53BP1 constructs were calculated by a Mann-Whitney test.
DOI: https://doi.org/10.7554/eLife.44353.012
The following figure supplement is available for figure 6:

**Figure supplement 1.** RAD9 - 53BP1 Proximity Ligation Assay in U2OS Cells.
DOI: https://doi.org/10.7554/eLife.44353.013

## Discussion

53BP1 is a highly post-translationally modified protein, with more than 200 phosphorylation sites documented in the human protein (*Hornbeck et al., 2004*), of which only a handful have been associated with any biological function (*Panier and Boulton, 2014*). This presents a truly enormous challenge to understanding which of these phosphorylations are significant in which aspect of the complex biology of this protein, and what interactions or allosteric changes they mediate.

Here we have identified a set of experimentally observed phosphorylation sites that mediate interaction of 53BP1 with TOPBP1 and thereby facilitate important aspects of the DNA damage response, mirroring the behaviour of their fission and budding yeast homologues Crb2 and Rad9p, and Rad4 and Dpb11. The major interacting sites, Ser366 and Thr670, are strongly conserved in metazoa, and found to be phosphorylated in human, mouse and rat cells (*Hornbeck et al., 2004*). Although these SP/TP sites resemble typical CDK substrates, we find that their phosphorylation is enhanced in response to DNA damage, and is not prevented by a broad range of CDK kinase inhibitors (data not shown) suggesting that they are not under cell cycle control. This contrasts with fission and budding yeasts, where phosphorylation of the sites mediating Crb2 and Rad9p interaction with Rad4 and Dpb11p are under CDK-dependent cell cycle regulation (*Du et al., 2006*; *Pfander and Diffley, 2011*; *Qu et al., 2013*). However, a recent study also identifies a DNA damage-dependent and cell cycle-independent pathway for phosphorylation of the sites mediating interaction of budding yeast Rad9p with Dpb11p (*di Cicco et al., 2017*), which may be comparable to that we describe here for the human proteins.

Although mutation of either Ser366 or Thr670 impacts on the DNA damage response, the strongest effects are seen when neither site can be phosphorylated. This suggests that the affinity of TOPBP1 for 53BP1 is enhanced by cooperative binding of BRCT1,2 and BRCT4,5 of TOPBP1 to a single pS366-pS670-53BP1 molecule, and may explain the failure of isolated TOPBP1-BRCT0,1,2 and BRCT4,5 constructs to co-localise with 53BP1 at sites of DNA damage (*Cescutti et al., 2010*).

While disruption of the TOPBP1-53BP1 interaction had no effect on co-localisation of pATM with 53BP1 in G1, it severely disrupts co-localisation of ATR with 53BP1, downstream activation of CHK1 and TP53, and consequent G1/S checkpoint. This is fully consistent with observations of the importance of ATR for repair of IR-induced damage in G1 (*Gamper et al., 2013*), as well as its well-described role in S and G2. The prevailing model for ATR checkpoint activation in late S and G2 phases, depends on the presence of extended segments of ssDNA and is usually associated with repair by homologous recombination (HR). DNA double-strand breaks (DSB) occurring in G1, where homologous sister chromatids are not available, are preferentially repaired via NHEJ rather than HR. However, recent evidence suggests that a limited amount of resection nonetheless occurs at G1 DSBs as a prelude to canonical NHEJ repair (*Averbeck et al., 2014*; *Biehs et al., 2017*) and generates sufficient ssDNA to bind at least a few molecules of RPA, providing a platform for recruiting ATR-ATRIP. In these circumstances, rather than RAD17-RFC and 9-1-1 then being sufficient for bringing in TOPBP1 to fully anchor and activate ATR-ATRIP, our data strongly suggest that it is instead 53BP1 that plays this critical role in G1, through interaction of DNA damage responsive phosphorylation sites at Ser366 and Thr670, with the BRCT-domain clusters in the N-terminus of TOPBP1 (*Figure 7*).

We previously showed that TOPBP1-BRCT1 provides the highest affinity binding-site for pRAD9 (*Rappas et al., 2011*; *Day et al., 2018*), and the data presented here implicate BRCT2 and BRCT5

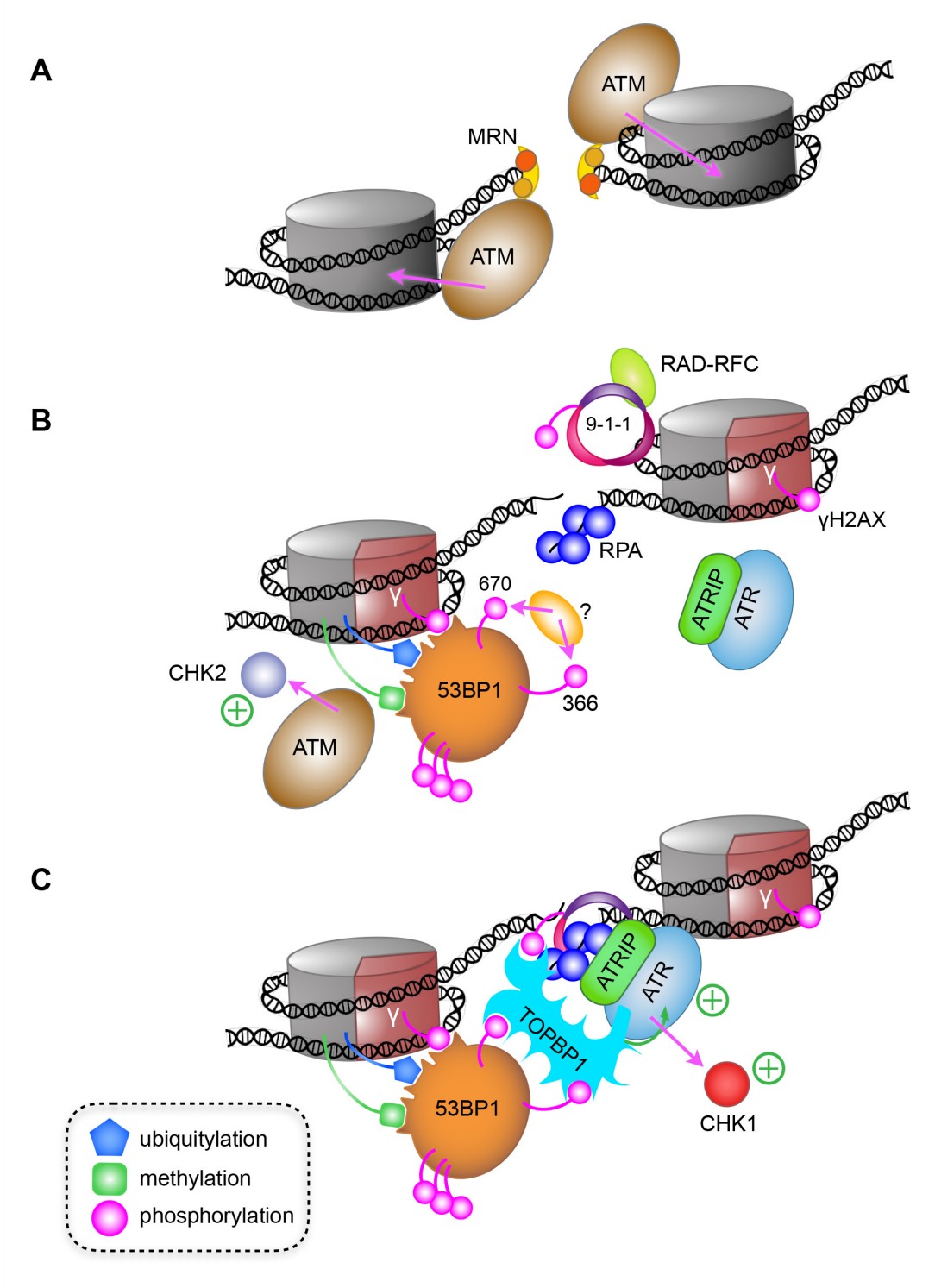

**Figure 7.** A model for ATR activation through phospho-dependent interaction of 53BP1 and TOPBP1. (**A**) Following irradiation, the Mre11-Rad50-Nbs1 (MRN) complex is recruited to broken ends of a DNA double-strand break, and facilitates recruitment and activation of ATM, which phosphorylates H2AX-Ser139 to generate the γ-H2AX signal. (**B**) Limited resection of the broken ends by MRN and CtIP (not shown), provides binding sites for the ssDNA-binding RPA complex, and for loading of the RAD9-RAD1-HUS1 checkpoint clamp (9-1-1) at the dsDNA-ssDNA junction by the RAD-RFC clamp loader. The γ-H2AX signal leads to recruitment of MDC1 and RNF168 (not shown) resulting in H2A ubiquitylation and consequent recruitment of 53BP1, which interacts with multiple post-translational modifications on nucleosomes in the vicinity of the break. (**C**) Phosphorylation of 53BP1-Ser366 and

*Figure 7 continued on next page*

*Figure 7 continued*

Thr670 by an as yet unidentified kinase, facilitates 53BP1 interaction with TOPBP1, which can simultaneously bind 9-1-1 via the phosphorylated C-terminus of RAD9, leading to recruitment and activation of ATR and CHK1. Whether the 53BP1 and ATR-ATRIP complexes bridged by TOPBP1 are on the same side of a break, or on opposite sides of a break as shown here, remains to be determined.

DOI: https://doi.org/10.7554/eLife.44353.014

as the primary binding sites for phosphorylated 53BP1. Thus, a single TOPBP1 molecule could in principle bind 53BP1 and 9-1-1 in concert. In support of this idea we found that DNA damage causes both of these TOPBP1 ligands to localise not more than 30–40 nm from each other in DNA damage foci – a distance comparable to ~100 bp of linear DNA or the predicted length of the unstructured tail of RAD9 in a random-coil conformation – making it highly likely that the 53BP1 and RAD9 proteins generating the PLA signals are bound to the same TOPBP1 molecule. Regardless of this involvement of 9-1-1, our data clearly show that it is the interaction of TOPBP1 with 53BP1 that is critical for ATR localisation and activation in G1. These results reveal a consistent role for TOPBP1 in localising and activating ATR at sites of DNA damage, but show that this depends on phosphorylation-mediated interactions with different recruitment factors at different stages of the cell cycle.

# Materials and methods

## Key resources table

| Reagent type (species) or resource | Designation | Source or reference | Identifiers | Additional information |
|---|---|---|---|---|
| Strain, strain background | *E. coli* BL21(DE3) | New England Biolabs | Cat#C2527 | |
| Cell line (*Homo sapiens*) | Hela cells | ATCC | RCB0007 | |
| Cell line (*Homo sapiens*) | U2OS cells | STRV | | |
| Cell line (*Homo sapiens*) | U2OS cells stably expressing eYFP-53BP1 WT, S366A and T670A | This study | | |
| Cell line (*Homo sapiens*) | RPE-1 htert WT | STRV | | |
| Transfected construct | peYFP-53BP1 WT, peYFP-53BP1 S366A, peYFP-53BP1 T670A, peYFP-53BP1 S366A T670A | This study | | |
| Antibody | Mouse monoclonal anti-GFP [LGB-1] | Abcam | Cat#ab291, RRID:AB_449092 | (1:100) dilution IF |
| Antibody | Rabbit monoclonal anti-pATM (S1981) | Abcam | Cat#2152–1 RRID:AB_991678 lot GR217573-12 | (1:100) dilution IF |
| Antibody | Donkey polyclonal anti-goat Alexa Fluor 647 | Abcam | Cat#ab150135 RRID:AB_2687955 | (1:400) dilution IF |
| Antibody | Goat polyclonal anti-53BP1 | Bethyl Laboratories, Inc | Cat#A303-906A RRID:AB_2620256 | (1:100) dilution IF |
| Antibody | Rabbit polyclonal anti-53BP1 | Bethyl Laboratories, Inc | Cat#A300-272A RRID:AB_185520 | (1:100) dilution IF |
| Antibody | Rabbit polyclonal anti-Rad9 | Bethyl Laboratories, Inc | Cat#A300-890A RRID:AB_2269209 | (1:100) dilution IF |
| Antibody | Rabbit polyclonal anti-TopBP1 | Bethyl Laboratories, Inc | Cat#A300-111A RRID:AB_2272050 | (1:50) dilution IF, (1:500) dilution WB |

*Continued on next page*

*Continued*

| Reagent type (species) or resource | Designation | Source or reference | Identifiers | Additional information |
|---|---|---|---|---|
| Antibody | Rabbit polyclonal anti-β-actin (13E5) | Cell Signaling Technology | Cat#4970 also 4970P, 4970L,4970S RRID:AB_2223172 lot 14 | (1:1000) dilution WB |
| Antibody | Mouse monoclonal anti-p53 (DO-7) | Cell Signaling Technology | Cat# 48818 RRID:AB_2713958 lot 1 | (1:100) dilution IF |
| Antibody | Rabbit polyclonal anti-p-p53 (S15) | Cell Signaling Technology | Cat#9284also9284S, 9284L,9284P RRID:AB_331464 lot 19 | (1:100) dilution IF |
| Antibody | Rabbit monoclonal anti-p21 Waf1/Cip1 (12D1) | Cell Signaling Technology | Cat#2947also 2947S,2947P RRID:AB_823586 lot 9 | (1:100) dilution IF |
| Antibody | Goat polyclonal anti-mouse IgG Fab2 Alexa Fluor 647 Molecular Probes | Cell Signaling Technology | Cat#4410S RRID: AB_10694714 lot 11 | (1:400) dilution IF |
| Antibody | Goat anti-rabbit IgG, HRP linked Antibody | Cell Signaling Technology | Cat#7074also7074S, 7074V,7074P2 RRID:AB_2099233 lot 26 | (1:2000) dilution WB |
| Antibody | Goat anti-mouse IgG, HRP linked Antibody | Cell Signaling Technology | Cat#7076also7076S, 7076V,7076P2 RRID:AB_330924 lot 32 | (1:2000) dilution WB |
| Antibody | Mouse monoclonal anti-53BP1 | EMD Millipore Corp | Cat#MAB3802 RRID:AB_2206767 lot 2794909 | (1:500) dilution IF, (1:500) dilution WB |
| Antibody | Rabbit polyclonal anti-pATR (pT1989) | GeneTex | Cat#GTX128145 RRID:AB_2687562 | (1:100) dilution IF, (1:500) dilution WB |
| Antibody | Rabbit polyclonal anti-pS366 53BP1 | ImmunoKontact | Antigen:[CSSDLVAP (pS)PDAFRSTP] | (1:500) dilution IF, (1:500) dilution WB |
| Antibody | Rabbit polyclonal anti-pT670 53BP1 | ImmunoKontact | Antigen:[CVEEIPE (pT)PCESQGEE] | (1:500) dilution IF, (1:500) dilution WB |
| Antibody | Mouse monoclonal anti-BrdU Monoclonal Antibody (MoBU-1), Alexa Fluor 488 | Invitrogen by ThermoFisher Scientific | Cat#B35130 RRID:AB_2536434 lot 1712859 | (1:200) dilution IF |
| Antibody | Donkey polyclonal anti-rabbit DKXRB TRITC | Invitrogen by ThermoFisher Scientific | Cat#A16040 RRID:AB_2534714 lot 31-33-091912 | (1:400) dilution IF |
| Antibody | Donkey polyclonal anti-mouse DKXMU IgG F(AB)' 2 FITC | Invitrogen by ThermoFisher Scientific | Cat#A24507 RRID: AB_2535976 lot 42-73-052014 | (1:400) dilution IF |
| Antibody | Goat polyclonal anti-rabbit IgG (H + L) Cross-Adsorbed Goat Secondary Antibody, Cyanine5 | Invitrogen by ThermoFisher Scientific | Cat#A10523 RRID: AB_10374302 lot 1675037 | (1:400) dilution IF |
| Antibody | Mouse monoclonal anti-Cyclin A (B-8) | Santa Cruz Biotechnology | Cat#sc-271682 RRID: AB_10709300 lot L1316 | (1:100) dilution IF |
| Antibody | Goat polyclonal anti-ATR (N-19) | Santa Cruz Biotechnology | Cat#sc-1887 RRID: AB_630893 lot G1408 | (1:500) dilution WB |
| Antibody | Rabbit polyclonal anti-HA-probe (Y-11) | Santa Cruz Biotechnology | Cat#sc-805 RRID: AB_631618 lot C0415 | (1:100) dilution IF, (1:2000) dilution WB |
| Antibody | Mouse monoclonal anti-HA-probe (F-7) | Santa Cruz Biotechnology | Cat#sc-7392 RRID: AB_627809 lot C1114 | (1:100) dilution IF |
| Antibody | Rabbit polyclonal anti-Tubulin (H-235) | Santa Cruz Biotechnology | Cat#sc-9104 RRID: AB_2241191 lot L1713 | (1:2000) dilution WB |

*Continued on next page*

*Continued*

| Reagent type (species) or resource | Designation | Source or reference | Identifiers | Additional information |
|---|---|---|---|---|
| Recombinant DNA reagent | Plasmid: pCMH6K HA-53BP1 | *Noon et al., 2010* a gift from Penny Jeggo | N/A | Plasmid encoding full length Human 53BP1 WT, S366A, T670A or S366A T670A mutants. Contains silent mutations for siRNA resistance. |
| Recombinant DNA reagent | Plasmid: peYFP-53BP1 | This paper | N/A | Plasmid encoding full length Human 53BP1 WT, S366A, T670A or S366A T670A mutants. Contains silent mutations for siRNA resistance. |
| Recombinant DNA reagent | Plasmid: peYFP-C1 | | N/A | |
| Sequence-based reagent | siRNA targeting sequence: SCR siRNA: sense: UUCAAUAAAU UCUUGAGGU(dTdT) antisense: (dTdT) CCTCAAGAATTTATTGAA | Eurofins (*Lou et al., 2003*) | | |
| Sequence-based reagent | siRNA targeting sequence: 53BP1 siRNA*: sense: AGAACGAGGAGA CGGUAAUAG UGGG(dTdT) antisense: (dTdT)CCCACTATT ACCGTCTCCTCGTTCT | Eurofins (*Noon et al., 2010*) | | |
| Sequence-based reagent | siRNA targeting sequence: 3' UTR 53BP1 siRNA**: sense: AAAUGUGUCUU GUGUGUAA(dTdT) antisense: (dTdT)TTACACA CAAGACACATTT | Eurofins (*Knobel et al., 2014*) | | |
| Sequence-based reagent | siRNA targeting sequence: TOPBP1 : sense: GUAAAUAUCU GAAGCUGUA(dTdT) antisense: (dTdT) UACAGC UUCAGAUAUUUAC | Eurofins (*Broderick et al., 2015*) | | |
| Sequence-based reagent | siRNA targeting: ATR siRNA ID: s536 | ThermoFisher Scientific | | |
| Sequence-based reagent | Primer: 53BP1 cloning fragment 1 Forward (5'- > 3'): GTCCGGA CTCAGATCTAT GGACCCTACTG GAAGTCAGT | Eurofins (this paper) | | Primer used for PCR in cloning experiment |
| Sequence-based reagent | Primer: 53BP1 cloning fragment 1 Reverse (5'- > 3'): CACACTGGCGTCCCT GTCTGACTGACC | Eurofins (this paper) | | Primer used for PCR in cloning experiment |

*Continued on next page*

Continued

| Reagent type (species) or resource | Designation | Source or reference | Identifiers | Additional information |
|---|---|---|---|---|
| Sequence-based reagent | Primer: 53BP1 cloning fragment 2 Forward (5'- > 3'): AGGGACGCCAGTG TGTGAGGAGGATGGT | Eurofins (this paper) | | Primer used for PCR in cloning experiment |
| Sequence-based reagent | Primer: 53BP1 cloning fragment 2 Reverse (5'- > 3'): TAGATCCGGT GGATCCTTAGTGA GAAACATAATCGT GTTTATATTTTGGATGCT | Eurofins (this paper) | | Primer used for PCR in cloning experiment |
| Sequence-based reagent | Primer: 53BP1 S366A mutagenesis Forward (5'- > 3'): TTGTTGCTCC tgcTCCTGATGCT | Eurofins (this paper) | | Primer used for PCR in cloning experiment |
| Sequence-based reagent | Primer: 53BP1 S366A mutagenesis Reverse (5'- > 3'): GATCTGAAGAA TTCGTGGAAAGAC | Eurofins (this paper) | | Primer used for PCR in cloning experiment |
| Sequence-based reagent | Primer: 53BP1 T670A mutagenesis Forward (5'- > 3'): AATCCCTGA GgcaCCTTGTGAAAG | Eurofins (this paper) | | Primer used for PCR in cloning experiment |
| Sequence-based reagent | Primer: 53BP1 T670A mutagenesis Reverse (5'- > 3'): TCTTCCA CCTCAGACCCTG | Eurofins (this paper) | | Primer used for PCR in cloning experiment |
| Peptide, recombinant protein | 53BP1 pT334 peptide 'Flu'-GYGGGC SLAS(pT)PATTLHL | Peptide Protein Research Limited (this paper) | | Fluorescein labelled for FP measurements |
| Peptide, recombinant protein | 53BP1 pS366 peptide 'Flu'-GYGSSDLVAP (pS)PDAFRST | Peptide Protein Research Limited (this paper) | | Fluorescein labelled for FP measurements |
| Peptide, recombinant protein | 53BP1 pS380 peptide 'Flu'-GYGTPFIVPS (pS)PTEQEGR | Peptide Protein Research Limited (this paper) | | Fluorescein labelled for FP measurements |
| Peptide, recombinant protein | 53BP1 pT670 peptide 'Flu'- GYGEVEEIPE(pT) PCESQGE | Peptide Protein Research Limited (this paper) | | Fluorescein labelled for FP measurements |
| Peptide, recombinant protein | 53BP1 pS366 peptide SSDLVAP(pS)PDAFRST | Peptide Protein Research Limited (this paper) | | |
| Peptide, recombinant protein | 53BP1 pT670 peptide EVEEIPE(pT)PCESQGE | Peptide Protein Research Limited (this paper) | | |
| Commercial assay or kit | In-Fusion HD Cloning Kit | Clonetech | Cat#639646 | |
| Commercial assay or kit | APEX Alexa Fluor 555 Antibody Labeling Kit (used for pS366 and pT670 53BP1 antibodies) | Invitrogen by ThermoFisher Scientific | Cat#A10470 lot 1831224 | |

Continued

| Reagent type (species) or resource | Designation | Source or reference | Identifiers | Additional information |
|---|---|---|---|---|
| Commercial assay or kit | Click-iT EdU Alexa Fluor 647 Imaging Kit | Invitrogen by ThermoFisher Scientific | Cat#C10340 | |
| Commercial assay or kit | Q5 Site-Directed Mutagenesis Kit | New England Biolabs | Cat#E0554S | |
| Commercial assay or kit | Premo FUCCI Cell Cycle Sensor (BacMam 2.0) | ThermoFisher Scientific | Cat#P36238 | |
| Commercial assay or kit | Pierce ECL Western Blotting Substrate | ThermoFisher Scientific | Cat#32209 lot RE232713 | |
| Commercial assay or kit | Cell Line Nucleofector Kit V | Lonza | Cat#VCA-1003 | |
| Chemical compound, drug | NanoJuice Transfection Kit | EMD Millipore Corp | Cat#71902 | |
| Chemical compound, drug | Fisher BioReagents Bovine Serum Albumin (BSA) Fatty Acid-free Powder | Fisher Scientific by ThermoFisher Scientific | Cat# BP9704-100 CAS: 9048-46-8 | |
| Chemical compound, drug | ProLong Diamond Antifade Mountant ThermoFisher Scientific | Invitrogen by ThermoFisher Scientific | Cat# P36965 | |
| Chemical compound, drug | NuPAGE 3–8% Tris-Acetate Protein Gels | Invitrogen by ThermoFisher Scientific | Cat#EA0378BOX | |
| Chemical compound, drug | NuPAGE Antioxidant | Invitrogen by ThermoFisher Scientific | Cat#NP0005 | |
| Chemical compound, drug | NuPAGE Sample Reducing Agent (10X) | Invitrogen by ThermoFisher Scientific | Cat#NP0004 | |
| Chemical compound, drug | NuPAGE LDS Sample Buffer (4X) | Invitrogen by ThermoFisher Scientific | Cat#NP0007 | |
| Chemical compound, drug | Benzonase Nuclease | Santa Cruz Biotechnology | Cat#sc-202391 | |
| Chemical compound, drug | Phosphatase Inhibitor Cocktail C | Santa Cruz Biotechnology | Cat#sc-45065 | |
| Chemical compound, drug | G418 Disulfat Salt | Sigma-Aldrich | A1720 ; CAS: 108321-42-2 | |
| Chemical compound, drug | Nocodazole | Sigma-Aldrich | SML1665; CAS: 31430-18-9 | |
| Chemical compound, drug | 5-Bromo-2′-deoxyuridine (BrDU) | Sigma-Aldrich | B5002; CAS: 59-14-3 | |
| Chemical compound, drug | bisBenzimide H33352 trihydrochloride (Hoechst 33342) | Sigma-Aldrich | B2261 ; CAS: 23491-52-3 | |
| Chemical compound, drug | Monoclonal Anti-HA—Agarose antibody produced in mouse | Sigma-Aldrich | A2095 | |
| Chemical compound, drug | cOmplete, EDTA-free Protease Inhibitor Cocktail | Sigma-Aldrich | 000000005056489001; COEDTAF-RO ROCHE | |
| Chemical compound, drug | Duolink In Situ Orange Starter Kit Goat/Rabbit | Sigma-Aldrich | DUO92106 | |

*Continued*

| Reagent type (species) or resource | Designation | Source or reference | Identifiers | Additional information |
|---|---|---|---|---|
| Chemical compound, drug | Lipofectamine RNAiMAX Transfection Reagent | ThermoFisher Scientific | Cat# #13778015 | |
| Chemical compound, drug | Phusion Flash High Fidelity Master Mix | ThermoFisher Scientific | Cat#F-548 | |
| Chemical compound, drug | Pierce ECL Western Blotting Substrate | ThermoFisher Scientific | Cat#32209 lot RE232713 | |
| Software, algorithm | Prism six for Mac OS X (v6.0h) | GraphPad Software | https://www.graphpad.com RRID:SCR_002798 | |
| Software, algorithm | Cell Profiler (2.2.0) | Broad Institute | http://cellprofiler.org/ RRID:SCR_007358 | |
| Software, algorithm | FIJI | ImageJ software | http://fiji.sc/ RRID:SCR_002285 | |
| Software, algorithm | Micro-Manager (µManager) | Vale Lab, UCSF | https://micro-manager.org/ RRID:SCR_000415 | |
| Software, algorithm | SlideBook6 | Intelligent Imaging Innovations (3i) | https://www.intelligent-imaging.com/slidebook RRID:SCR_014300 | |
| Software, algorithm | SnapGene | GSL Biotech LLC | http://www.snapgene.com/ RRID:SCR_015052 | |
| Software, algorithm | NEBaseChanger v1.2.6 | New England Biolabs | http://nebasechanger.neb.com/ | |
| Software, algorithm | CCP4 | Combined Crystallographic Computing Project | http://www.ccp4.ac.uk/ RRID:SCR_007255 | |
| Software, algorithm | Phenix | Phenix Consortium | https://www.phenix-online.org/ RRID:SCR_014224 | |
| Software, algorithm | Buster | Global Phasing | https://www.globalphasing.com/buster/ RRID:SCR_015653 | |
| Other | Microscope: Olympus-3i spinning disc | Olympus | N/A | |
| Other | Microscope: Olympus IX70 Core DeltaVision | Olympus | N/A | |
| Other | Bioruptor Pico sonication device | Diagenode | Cat# B01060001 | |
| Other | ImageQuant LAS 4000 | GE Healthcare Life Sciences | Cat#28955810 | |

## Contact for reagent sharing

Further information and requests for reagents should be directed to and will be fulfilled by the Lead Contact, Laurence Pearl (laurence.pearl@sussex.ac.uk).

## Experimental model and subject details

See Key Resources Table for information on bacterial strains used as sources of material in this study.

## Generation of pEYFP-53BP1

Two separate fragments of 53BP1 cDNA were amplified by PCR from the pCMH6K HA-53BP1 plasmid harbouring three silent mutations that confer siRNA resistance (Noon et al., 2010). The two amplicons were inserted into the BglII/BamHI sites of peYFP-C1a by In-Fusion cloning (Clontech). Ser366Ala (S366A), Thr670Ala (T670A) and Ser366Ala/Thr670Ala (S366A/T670A) mutations were created by site-directed mutagenesis. All constructs were verified by Sanger sequencing.

## Cell culture and transfection

HeLa, U2OS and RPE1 cells used in this study derive from central stocks in the Genome Damage and Stability Center at the University of Sussex (http://www.sussex.ac.uk/gdsc/facilities) and are STR-validated and determined as mycoplasma free by MycoAlert (Lonza). HeLa and U2OS cells were cultured in DMEM supplemented with 10% (v/v) fetal calf serum (FCS), 1% (v/v) penicillin/streptomycin mix, and 1% (v/v) L-glutamine. RPE1 active hTert cells were cultured in DMEM F12 supplemented with 10% (v/v) FCS and 1% (v/v) penicillin/streptomycin mix. Transfections with siRNA were carried out with Lipofectamine RNAiMAX Transfection Reagent (ThermoFisher Scientific). Briefly, $0.25 \times 10^6$ cells were seeded in 35 mm wells. The next day, according to the manufacturer's instructions, 7.5 µL per well of RNAiMAX were used to transfect cells, obtaining a final concentration of 20 nM for the added siRNA (53BP1, ATR, TOPBP1, see Key Resources Table for sequences). Knockdown efficiencies were confirmed either by western blot or immunofluorescence. All experiments requiring prior siRNA transfection, involved an 72 hr period of exposure to siRNA treatment.

Complementation experiments were performed by transiently transfecting siRNA-resistant 53BP1 constructs (pCMH6K HA-53BP1 or pEYFP-53BP1 depending on the experiment). For each 35 mm well containing HeLa cells, 1.5 µg of plasmid was transfected with NanoJuice Core Transfection Reagent and Booster at a ratio of 3:1 (Reagent:DNA). U2OS and RPE1 cells were transfected using Cell Line Nucleofector Kit V and a Nucleofector electroporator (Lonza) according to the manufacturer's instructions. Briefly, $1 \times 10^6$ cells were electroporated with 1.5 µg of plasmid, then allowed to recover for a period of 6 hr in their respective 20% FCS media. The media was then refreshed (10% FCS), and cells cultured for a further 14 hr to allow protein expression from the transfected plasmids.

## Generation of stable U2OS cell lines

Stably transfected U2OS cells were generated using using Cell Line Nucleofector Kit V and a Nucleofector electroporator (Lonza). After 24 hr, cells that had integrated plasmid were selected in media supplemented with G418 (400 µg mL$^{-1}$ Sigma-Aldrich) for 10 days. Surviving colonies were used to seed subsequent experiments.

## Method details

### Protein expression and purification

TOPBP1 constructs for biochemical and structural analysis were expressed in *E. coli* and purified by conventional chromatography, as previously described (Rappas et al., 2011; Qu et al., 2013; Day et al., 2018).

### Fluorescence polarization experiments

Binding to TOPBP1-BRCT0,1,2, and -BRCT4,5 domains were determined using fluorescein-labelled peptides and BRCT fusion proteins as previously described (Rappas et al., 2011; Qu et al., 2013; Day et al., 2018).

### X-ray crystallography

Co-crystals of the TOPBP1-BRCT0,1,2–53BP1-pT670 complex and TOPBP1-BRCT4,5–53BP1-pS366 complex were grown by vapour diffusion from conditions optimised from initial hits in E1 (10% w/v PEG 20 000, 20% v/v PEG MME 550 0.03 M of each ethylene glycol 0.1 M MES/imidazole pH 6.5) and G7 (10% w/v PEG 4000, 20% v/v glycerol 0.02 M of each carboxylic acid 0.1 M MOPS/HEPES-Na pH 7.5) of the MORPHEUS screen (Molecular Dimensions) prior to plunge-freezing in liquid nitrogen. Data were collected on the I04 and I03 beamlines at the Diamond Synchrotron Lightsource and the structures were determined by molecular replacement using PDB models 2XNH and 3UEN.

Processing and refinement were carried out using the CCP4 and PHENIX suites of programs. For the final TOPBP1-BRCT4,5–53BP1-pS366 structure refinements with Phenix, NCS was imposed with chains A and C, B and D, and P and R, being paired together. Statistics for data collection and refinement are presented in *Supplementary file 1*. Coordinates and structure factors have been deposited in the Protein Databank with accession codes **6RML** (pT670 complex) and **6RMM** (pS366 complex).

## DNA damage induction
Unless otherwise stated, DNA damage was produced by exposing cells to a 9 Gy radiation dose using a Gamma-cell 1000 Elite irradiator (Caesium137 gamma source). Post-exposure, cells were allowed to recover for a period of 4 hr at 37°C.

## G1/S cell cycle detection
Prior to infection, $1 \times 10^5$ eYFP-53BP1 WT Ser366Ala or Thr670Ala U2OS cells were first transfected in 35 mm wells with 53BP1 siRNA. Twenty-four hours later, cells were incubated with ~20 viral particles per cell (20 µL) of Cdt1-RFP Premo FUCCI Cell Cycle Sensor (ThermoFisher Scientific). After a further period of 24 hr, cells were exposed to gamma radiation.

## Immunofluorescence and microscopy
Transfected or non-transfected cells were cultured on 10 mm round cover glasses (VWR). Prior to fixation, cells were incubated with Hoechst 33342 (5 µg mL$^{-1}$ in PBS) for 15 min at 37°C, then washed three times with ice-cold PBS. Cells were fixed with cold methanol for 20 min (−20°C) (unless otherwise stated) and again washed three times with ice-cold PBS. Cells were blocked using a 4% (w/v) bovine serum albumin (BSA) solution in PBS for 15 min on ice. Primary and secondary antibody incubations were carried out at room temperature for 1 hr in BSA/PBS, followed by three sequential washes with PBS and a single wash with 0.1% (v/v) TritonX-100 in PBS. Mounting on glass slides was carried out with ProLong Diamond Antifade Mountant (ThermoFisher Scientific).

Images were acquired with either an Olympus-3i spinning disc microscope equipped with a Hamamatsu ORCA-flash4.0lT digital CMOS camera and using an UPLANSAPO 60X/1.35 oil objective in confocal mode, or with an Olympus IX70 Core DeltaVision microscope equipped with a CoolSnap HQ$^2$ camera and an UApo N340 40X/1.35 oil immersion objective. Images were captured sequentially at designated wavelengths at a resolution of 512 × 512 pixels. Scale bars were added to pictures after calculation to convert pixels to micrometres (µm).

Images were analysed using CellProfiler (http://cellprofiler.org) (*Carpenter et al., 2006*). Nuclei were detected as primary objects, with foci detection and mean intensity fluorescence measurement carried out for each object.

The quoted correlation coefficient, ranges from −1 (complete inverse correlation) to +1 (complete correlation), and corresponds to the measured normalised covariance (covariance divided by the product of standard deviation of pixels in each image) similar to a Pearson's coefficient. Montages of representative pictures were created using FIJI (http://fiji.sc) (*Preibisch et al., 2009*).

## Checkpoint analysis
U2OS cells were reverse-transfected with 53BP1 siRNA. Forty-eight hours later, cells were transfected with an siRNA-resistant construct containing either wild-type HA-53BP1 or one of the phosphorylation mutants. Fifteen hours later, cells were incubated for a period of 1 hr with EdU (10 µM) and irradiated with 2 Gy ionising radiation, before the addition of BrdU (10 µM) and nocodazole (0.25 µg mL$^{-1}$). Cells were incubated for a further period of 7 hr before staining. Fixation of cells was performed using a 4% (v/v) paraformaldehyde in PBS solution at room temperature for 15 min. Fixed cells were then permeabilised with a 0.5% (v/v) TritonX-100 in PBS solution for 20 min at room temperature. The subsequent steps for EdU staining carried out as indicated in the protocol provided with the Click-iT EdU Alexa Fluor 647 Imaging Kit (ThermoFisher Scientific).

Detection of BrdU required prior incubation of cells with 2N HCl for 60 min at 37°C in order to denature genomic DNA. After neutralisation of the HCl with boric acid (pH8.5) for a period of 30 min, cells were then blocked with a 4% (w/v) BSA/PBS solution for 1 hr. Antibody-staining for HA and BrdU were performed overnight at 4°C. Treated cells were imaged with a ScanR microscope

(Olympus Life Science), with cells staining for EdU incorporation ignored for subsequent analysis. The entry of cells into S-phase, after DNA damage was quantified by counting only cells that had incorporated BrdU, but not EdU (BrdU+/EdU-).

To determine the proportion of cells at each cell cycle phase, U2OS cells processed as above, were also analysed on the ScanR microscope, after 4 hr of exposure to 8 Gy of ionising radiation. Only cells expressing either wild-type HA-53BP1 or phosphorylation site mutants (Ser366Ala and Thr670Ala) were compared. Thirty-six images per well were analysed for DAPI intensity and total internal FITC intensity, in order to generate a cell cycle profile of transfected cells.

## Western blots

For western blots of damage-induced 53BP1 phosphorylation, HeLa cells were reverse transfected with either siRNA targeting 53BP1 or a non-targeting control, and then incubated for a period of 48 hr at 37 °C. Cells were then exposed to 8 Gy of ionising radiation and allowed to recover for 4 hr before being lysed by re-suspension of the frozen cell pellets in 2 mL of RIPA buffer (Sigma-Aldrich) supplemented with EDTA-free protease- and PhosSTOP phosphatase-inhibitor tablets (Roche Diagnostics, Burgess Hill, UK) and 40 µl Benzonase endonuclease (25 Units $\mu L^{-1}$, Merck-Millipore), followed by disruption in a Bioruptor Pico with water cooler (Diagenode, Seraing, Belgium). Cell debris and insoluble material were removed by centrifugation at 16,000 x $g$, for 10 min at 4°C, followed by dilution in 10 mM Tris-HCl pH 7.5, 150 mM NaCl, 0.5 mM EDTA, 0.5 mM TCEP supplemented with protease and phosphatase inhibitor tablets as before.

Following separation by SDS-PAGE, samples were transferred to a nitrocellulose membrane and probed for the presence of immuno-reactive species by chemi-luminescent western blot (see Key Resources Tables for details of antibodies and dilutions).

## Proximity Ligation Assay (PLA)

For endogenous 53BP1-RAD9 protein proximity ligation assays, U2OS or RPE1 cells were seeded at a density of $2.5 \times 10^5$ cells per $cm^2$ on 10 mm round glass coverslips and cultured for 24 hr. Cells were irradiated (9Gy) or not (control), stained in a Hoechst 33342 (5 µg $mL^{-1}$ in PBS) solution and fixated with methanol for 20 min at −20°C. After three washes in PBS 1X, the ligation experiment was performed according to the manufacturer's instructions using the Duolink In Situ Orange Starter Kit Goat/Rabbit (Sigma-Aldrich). Briefly, cells were blocked for 60 min at 37°C in a heated humidified chamber and subsequently incubated with 53BP1 and Rad9 antibodies (Bethyl Laboratories) for 1 hr. After two washes, coverslips were incubated with a PLUS-MINUS probe solution for another 1 hr at 37°C followed by washes and a 30 min ligation step at 37°C. Eventually, proximity ligation events were amplified for 100 min at 37°C. After washes and mounting of the coverslips, proximity events were observed by fluorescence microscopy and normalised to the number of nuclei. In case of TOPBP1 knockdown experiment, transfections were carried out with Lipofectamine RNAiMAX Transfection Reagent (ThermoFisher Scientific). Briefly, $0.25 \times 10^6$ U2OS or RPE1 cells were seeded on 10 mm round glass coverslips in a 35 mm well. The next day, according to the manufacturer's instructions, 7.5 µL per well of RNAiMAX were used to transfect cells, obtaining a final concentration of 20 nM for the added TOPBP1 siRNA (see Key Resources Table for sequences). After an 72 hr period of exposure to TOPBP1 siRNA, samples were submitted to the PLA.

For the transfected eYFP-53BP1 – RAD9 proximity ligation assays, U2OS cells knocked-down for endogenous 53BP1 were transfected with eYFP-53BP1 WT or the double mutant S366A T670A. Cells were irradiated (9 Gy) and stained with Hoechst 33342 after a 3 hr recovery period. Subsequently cells were fixated at −20°C with methanol for 20 min. A proximity ligation assay (PLA) was performed using the Duolink In Situ Detection Kit (Sigma Aldrich) according to the manufacturer's instructions, using an anti-Rad9 antibody (Rabbit polyclonal, A300-890A-T, Bethyl Laboratories, Inc) and an anti-GFP antibody (Mouse monoclonal [LGB-1], Abcam) to respectively detect the endogenous Rad9 and eYFP-53BP1. Several Z stacks including the depth of nuclei were acquired using an Olympus-3i spinning disc microscope in confocal mode. Images were Z projected and PLA events were analysed with Cell Profiler. More than 200 nuclei were counted per case. Results are represented as boxplots showing the median, the mean and the 10th- 90th percentiles. Statistical significance was determined with a Mann-Whitney test. Scale bar: 10 µm.

## Quantification and statistical analysis

All statistical analysis was carried out with Prism six software (GraphPad Software Inc, CA USA). Dissociation constants ($K_d$) were determined by non-linear regression to a one site-specific binding model.

The statistical significance of changes in cell cycle was determined using a standard $\chi^2$-test.

When only two variables were compared, significance was assessed by a two-sided Student's t-test. When more than two variables were compared, significance was assessed by a non-parametric Kruskal-Wallis test, corrected by the Dunn's multiple comparison test.

Graphs show adjusted p-values only when differences are considered to be significant; $*p<0.05$; $**p<0.01$; $***p<0.001$. Histograms show mean values, with error bars corresponding to one standard deviation. Boxplots show median (bar), mean (cross), 10th and 90th percentiles, with outliers plotted individually.

## Data and software availability

Coordinates and structure factors for the TOPBP1-BRCT0,1,2 complex with 53BP1-pThr670 peptide and for the TOPBP1-BRCT4,5 complex with 53BP1-pSer366 peptide, have been deposited in the RCBS Protein Databank with accession codes 6RML and 6RMM.

## Acknowledgements

We thank Mark Roe for assistance with X-ray data collection, and Tony Carr, Penny Jeggo, Manuel Stucki and Andy Blackford for useful discussion. We are grateful to the Diamond Light Source Ltd., Didcot, UK, for access to synchrotron radiation and to the Wellcome Trust for support for X-ray diffraction facilities at the University of Sussex. This work was supported by Cancer Research UK Programme Grants C302/A14532 and C302/A24386 (AWO and LHP).

## Additional information

### Funding

| Funder | Grant reference number | Author |
| --- | --- | --- |
| Cancer Research UK | C302/A14532 | Antony W Oliver Laurence H Pearl |
| Cancer Research UK | C302/A24386 | Antony W Oliver Laurence H Pearl |

The funders had no role in study design, data collection and interpretation, or the decision to submit the work for publication.

### Author contributions

Nicolas Bigot, Data curation, Software, Validation, Investigation, Visualization, Methodology, Writing—review and editing; Matthew Day, Data curation, Validation, Investigation, Visualization, Methodology, Writing—review and editing; Robert A Baldock, Investigation, Visualization, Methodology, Writing—review and editing; Felicity Z Watts, Supervision, Investigation, Methodology, Writing—review and editing; Antony W Oliver, Conceptualization, Data curation, Supervision, Funding acquisition, Validation, Investigation, Visualization, Methodology, Project administration, Writing—review and editing; Laurence H Pearl, Conceptualization, Resources, Data curation, Supervision, Funding acquisition, Validation, Visualization, Methodology, Writing—original draft, Project administration, Writing—review and editing

### Author ORCIDs

Nicolas Bigot https://orcid.org/0000-0002-4247-0217
Matthew Day http://orcid.org/0000-0001-7218-867X
Robert A Baldock http://orcid.org/0000-0002-4649-2966

Antony W Oliver https://orcid.org/0000-0002-2912-8273
Laurence H Pearl https://orcid.org/0000-0002-6910-1809

**Decision letter and Author response**
Decision letter https://doi.org/10.7554/eLife.44353.023
Author response https://doi.org/10.7554/eLife.44353.024

## Additional files

### Supplementary files

• Source data 1. Underlying data for graphs/statistical analyses. *Figure 4B*, *Figure 4E*, *Figure 4—figure supplement 1C*, *Figure 4—figure supplement 1E*, *Figure 5B*, *Figure 5F*, *Figure 5H*, *Figure 5—figure supplement 1A*, *Figure 5—figure supplement 1B*, *Figure 5—figure supplement 1C*, *Figure 5—figure supplement 2C*, *Figure 6D*, *Figure 6E*, *Figure 6F*, *Figure 6G*, *Figure 6—figure supplement 1A*, *Figure 6—figure supplement 1B*, *Figure 6—figure supplement 1D*
DOI: https://doi.org/10.7554/eLife.44353.015

• Supplementary file 1. Crystallographic Data Collection and Refinement Statistics.
DOI: https://doi.org/10.7554/eLife.44353.016

• Transparent reporting form
DOI: https://doi.org/10.7554/eLife.44353.017

### Data availability

Atomic Coordinates and structure factors have been deposited in the Protein Databank under accession codes: 6RML and 6RMM.

The following datasets were generated:

| Author(s) | Year | Dataset title | Dataset URL | Database and Identifier |
|---|---|---|---|---|
| Bigot N, Day M, Baldock RA, Watts FZ, Oliver AW, Pearl LH | 2019 | Crystal structure of TOPBP1 BRCT0,1,2 in complex with a 53BP1 phosphopeptide | http://www.rcsb.org/structure/6RML | Protein Data Bank, 6RML |
| Bigot N, Day M, Baldock RA, Watts FZ, Oliver AW, Pearl LH | 2019 | Crystal structure of TOPBP1 BRCT4,5 in complex with a 53BP1 phosphopeptide | http://www.rcsb.org/structure/6RMM | Protein Data Bank, 6RMM |

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
