## [Decision Letter]

[Editors’ note: this article was originally rejected after discussions between the reviewers, but the authors were invited to resubmit after an appeal against the decision.]

Thank you for submitting your work entitled "Phosphorylation-mediated interactions with TOPBP1 couple 53BP1 and 9-1-1 to control the G1 DNA damage checkpoint" for consideration by *eLife*. Your article has been reviewed by three peer reviewers, including Volker Dötsch as the Reviewing Editor and Reviewer #1, and the evaluation has been overseen by a Senior Editor.

Our decision has been reached after consultation between the reviewers. Based on these discussions and the individual reviews below, we regret to inform you that your work will not be considered further for publication in *eLife*.

As you will see in the individual reviews, all reviewers recognize the importance of investigating the mechanistic details of checkpoint assembly and execution. The overall feeling, however, following discussion among the reviewers, was that significant aspects of this manuscript have been published previously, reducing the level of novelty. Some technical aspects of the checkpoint assays have also been questioned.

*Reviewer #1:*

Bigot et al. describe an investigation of the interaction of the two scaffold proteins TOPBP1 and 53BP1 with each other as well as with other proteins. Both proteins are major scaffold proteins that are important for organizing the DNA damage response. While TOPBP1 is involved in activating ATR following single strand breaks while 53BP1 is a key protein important for the decision as to which DNA repair mode of double strand breaks is being activated. In a recent study this group had identified consensus peptide sequences that in a phosphorylated mode are recognized by the different BRCT domains of TOPBP1. Building in these results the authors now show how two phosphorylation sites in 53BP1 are involved in mediating the interaction with TOPBP1 and that this interaction is important for the G1/S phase check point. They show that the activation of ATR is compromised when the 53BP1-TOPBP1 interaction is disrupted resulting in a reduced activation of p53.

Overall, this is a nice work that includes structural aspects of the described interactions as well as functional data based on cell culture experiments. These results are important for understanding the selective activation of ATR vs ATM.

The study presented is heavily based on co-localization to show interaction. It would be better to show direct interaction by IPs and the use of DNAse to investigate if interactions in coIP experiments are mediated by DNA or are direct. Have the authors tried to use IPs?

*Reviewer #2:*

This manuscript describes how phosphorylation at Ser366 and Thr670 of 53BP1 results in a phospho-dependent interaction with BRCT domains in TOPBP1. They determine the crystal structures for TOPBP1 BRCT domains with the two phosphopeptides of 53BP1 and generate phopho-specific antibodies. They then follow the role of this interaction upon DNA damage for recruitment of TOPBP1, ATR and CHK1 to 53BP1 foci, whereas it does not affect ATM recruitment. Loss of 53BP1 phosphorylation leads to a greatly reduced G1/S checkpoint and a perturbed cell cycle distribution. The authors then study the interaction with 9-1-1 complex, and show colocalization. They use a PLA assay to suggest that a single TOPBP1 molecule can bridge between 53BP1 and Rad9 in the 9-1-1 complex.

This is an interesting manuscript that goes significantly beyond what was known for the yeast homologs Rad4 and Crb. However, some further analysis is required to firm up the conclusions:

Specifically:

- It is not clear that the 9-1-1 colocalization is really phosphorylation dependent. This could be tested by a repeat the 9-1-1 colocalizations with the phospho-mutant vs WT 53BP1 in the knockdown context

- The PLA assay is interpreted as indicating that a single TOPBP1 molecule bridges between 53BP1 and Rad9. This could be tested in a complementation assay, where a BRCT1 and a BRCT2 mutant of TOPBP1 are simultaneously expressed.

- The crystal structures are relatively low resolution and not yet ideally refined. At this resolution (local) NCS restraints would do much to improve the quality of the final structures. The structures need to correct some local errors (e.g. an Arg side chain points out of the available density)

*Reviewer #3:*

This manuscript has multiple conceptual and technical weaknesses:

The interaction between 53BP1 and TopBP1 and its role in the G1 checkpoint has already been reported (Cescutti et al., 2010). Although this study provides some new details, the overall model is not significantly advanced. Furthermore, TopBP1 also interacts with MDC1 in a phosphorylation-mediated manner, which was also shown to regulate ATR activation (Wang et al. 2011 JCB). The structural details of the TopBP1-MDC1 interaction has been analyzed and reported (Leung et al., 2013). Given that the localization of 53BP1 to DNA damage sites is dependent on MDC1, the function of 53BP1 in ATR regulation must be regulated by MDC1. Overall, the results of this study only add new details to an existing model but do not advance the model significantly.

This manuscript also suffers from several significant technical problems. For example, many experiments in this study used Cyclin A- as a marker for G1 cells. Cyclin A is largely a soluble nuclear protein. However, this study apparently used detergent-extracted cells for Cyclin A staining (Figure 4A). I am not sure if they have identified G1 cells correctly.

The G1 checkpoint assay used in this study was not properly done. I don't see any BrdU+ EdU- cells in the si53BP1+IR sample in Figure 4C. I am not sure how the authors can conclude that 53BP1 knockdown abolished the G1 checkpoint. In Figure 4D, only BrdU labeling but not EdU/BrdU sequential labeling was done. This is not the proper G1 checkpoint assay. How can the authors be sure that these are G1 cells moving into S phase?

In Figure 3C, why did anti-pT670 antibody detect nuclear foci that did not colocalize with eYFP-53BP1?

In Figure 5A, pATR formation is not defective in cells expressing the TopBP1 T670A mutant. pATR just did not colocalize with eYFP-53BP1. Where is pATR in these cells? Would this result argue against the requirement of TopBP1 phosphorylation at T640 for ATR activation?

In Figure 5C, it is surprising that some many Chk1 foci were detected. Chk1 is mobile protein after DNA damage.

In Figure 6F, there are no controls to check the effects of siTopBP1 on 53BP1 and RAD9 foci. Both 53BP1 and RAD9 are known to localize to DNA damage sites through TopBP1-independnet mechanisms. How does TopBP1 affects the 53BP1-RAD9 colocalization?

The model in Figure 7 is not supported by data. What is the evidence that 53BP1 recruits TopBP1? If 53BP1 recruits TopBP1 in G1 cells, why are 53BP1 and RAD9 colocalized? What is the evidence that 53BP1 functions independently of RAD9 to regulate ATR? I don't understand the logic of the model proposed.

---

## [Author Response]

[Editors’ note: the author responses to the first round of peer review follow.]

Reviewer #1:[…] The study presented is heavily based on co-localization to show interaction. It would be better to show direct interaction by IPs and the use of DNAse to investigate if interactions in coIP experiments are mediated by DNA or are direct. Have the authors tried to use IPs?

We have based our study on microscopy techniques because we need to look at the interactions we are interested in as a function of the cell cycle stage, which we can determine in microscopy experiments on a per cell basis using markers such as Cdt1, CyclinA and/or the intensity of DAPI/Hoescht staining. Of course the resolution of standard or even super-resolution fluorescence microscopy does not mean that optically co-localising proteins are actually interacting, but we have extended the approach using the proximity ligation assay (PLA) which does allow us to assign a high probability of direct interaction while retaining the ability to assign the cell cycle stage of the cell in which the PLA signal was observed. To achieve the same in a biochemical approach, even if the IP could be made to work cleanly, would require us either to synchronise the cell cultures, which affects the signalling pathways we are looking at, or to try and sort cells using a FACS approach. This is not something we have any expertise in and we do not know how easy it is to FACS sort the cell types we use, which are quite adherent and better suited for microscopy than free suspension.

We have nonetheless performed some pull-down experiments early on in the study to look at the interaction of TOPBP1 and 53BP1 in asynchronous cell cultures, and did obtain some promising results. An example is shown in Author response image 1, in which we could obtain an immunodetectable signal for TOPBP1 co-precipitated from cells transfected with HA-tagged 53BP1 constructs. These data certainly show a lower yield of co-precipitated TOPBP1 from cells transfected with 53BP1 with mutations in the key phosphorylation sites, but the signals are not strong and don’t reliably exceed the likely variation in the transfection efficiency and expressions levels between the different 53BP1 constructs. Furthermore, the experiment was not always reproducible and so we took the decision not to include this data in the manuscript.

Large scaffold proteins such as 53BP1 and TOPBP1 make multiple interactions and are very intimately associated with chromatin – and not just with its DNA – so that the harsh treatment required to release them into the soluble fraction will commonly disrupt many of the protein-protein interactions in which we are interested. It would be possible in principle to cross-link the cells as is done in ChIP for example, and then to try and use mass spectrometry to look for cross-linked 53BP1 and TOPBP1 peptides. However, cross-link mass spec of whole cells is a very specialist activity and not something we could access within a reasonable time-scale.

Reviewer #2:[…] This is an interesting manuscript that goes significantly beyond what was known for the yeast homologs Rad4 and Crb. However, some further analysis is required to firm up the conclusions:Specifically:- It is not clear that the 9-1-1 colocalization is really phosphorylation dependent. This could be tested by a repeat the 9-1-1 colocalizations with the phospho-mutant vs WT 53BP1 in the knockdown context

We suggest that the observed co-localisation of 53BP1 and 9-1-1 results from their common and complementary ability to interact with TOPBP1 in a phospho-dependent manner. The phosphorylation-dependency of the 9-1-1 interaction with TOPBP1 is well documented in the literature while that of 53BP1 with TOPBP1 is clearly demonstrated here, but the reviewer is absolutely correct that we have not formally demonstrated that the co-localisation we observe for 53BP1 and 9-1-1 is fully dependent on the phosphorylation-dependent interaction of 53BP1 with TOPBP1, although this is a very reasonable conclusion to draw from the rest of the data, even without the experiment that the reviewer suggests. We have performed this additional experiment, by transfecting wild-type and S366A/T670A eYFP-53BP1 constructs into U2OS cells with siRNA knockdown of endogenous 53BP1, and observe a significant decrease on levels proximity ligation events between RAD9 and the eYFP-53BP1 that are fully consistent with the expectation from our model. This data has been added to Figure 6 in the revised manuscript. We thank the reviewer for their helpful suggestion which strengthens the conclusions of our work.

- The PLA assay is interpreted as indicating that a single TOPBP1 molecule bridges between 53BP1 and Rad9. This could be tested in a complementation assay, where a BRCT1 and a BRCT2 mutant of TOPBP1 are simultaneously expressed.

The Duolink PLA system we use is well established in the literature as reporting on protein-protein proximities with a maximal separation of 40nm – a distance about 1.5x the diameter of a single ribosome and comparable to the predicted contour length of the unstructured C-terminal tail of RAD9. On the basis of the TOPBP1-dependent PLA signal we observe between 9-1-1 and 53BP1, the simplest and most likely explanation is that they are binding to the same TOPBP1 molecule. However, we haven’t formally eliminated the possibility that TOPBP1 could be interacting with 9-1-1 and 53BP1 as a dimer (or higher oligomer) in which singly-liganded TOPBP1 molecules bring 9-1-1 and 53BP1 into close proximity through self-association.

There is a limited literature, all from the same lab, suggesting that mammalian TOPBP1 (but not its yeast homologues) can undergo oligomerisation dependent on phosphorylation of a residue in the ATR-activating domain by AKT, which then binds to the metazoan-specific C-terminal BRCT7,8 module of another TOPBP1 molecule. Significantly these authors show that this apparent oligomerisation switches TOPBP1 into a poorly described function in repression of the E2F1 transcription factor, and away from its role in the DNA damage checkpoint by inhibiting its recruitment to chromatin and ATR binding (Liu et al., 2013, Mol. Cell. Biol., 33:4685-4700). So, the only suggestion of native TOPBP1 oligomerisation that we are aware of in the literature, is explicitly not involved in the DNA damage checkpoint that its interaction with 9-1-1 and 53BP1 mediates.

Nonetheless, the elegant complementation experiment the reviewer suggests could certainly eliminate a TOPBP1 self-association model, even if it is unlikely, and would add support to our already very reasonable explanation of the TOPBP1-dependent PLA data. However, while this experiment is straightforward for the reviewer to suggest, it could take us several months to develop and implement. We currently have none of the constructs that would be required, and we would need to develop from scratch a stable ‘knockdown add-back’ U2OS or RPE1 cell line for TOPBP1 (complicated by the fact that TOPBP1 unlike 53BP1 is an essential gene), and simultaneously express similar levels of two different TOPBP1 mutants. Furthermore, TOPBP1 BRCT1 and BRCT2 are implicated in interactions with other proteins such as treslin/Sld3 (responsible for assembling the replicative CMG helicase), so disruption of these sites in vivo could have unpredictable consequences on the G1 DNA damage response that are independent of their interactions with 9-1-1 and 53BP1. While this experiment would certainly add to the manuscript, we would submit that it is not essential for the measured conclusion we make, that 9-1-1 and 53BP1 are brought into very close proximity at sites of DNA damage, and that this is most likely through interaction with the same TOPBP1.

- The crystal structures are relatively low resolution and not yet ideally refined. At this resolution (local) NCS restraints would do much to improve the quality of the final structures. The structures need to correct some local errors (e.g. an Arg side chain points out of the available density)

These are certainly not the best diffracting crystals in the world – especially those for the BRCT4,5 complex, but they are what we get and we were not able to improve the resolution beyond this. Nonetheless, the key interfaces between the BRCT domains and the bound phosphopeptides are well resolved and unambiguous, and the conclusions we draw from these structures are justified by the data. We did not apply NCS in the original refinement of BRCT45 as not all the copies in the asymmetric unit have bound peptide, and are therefore not strictly equivalent. However, the reviewer’s suggestion is a reasonable one and we have re-refined the structure as suggested using NCS restraints between the related pairs within the four copies in the asymmetric unit, giving a small improvement in Rfree, and have rectified sub-optimal rotamers.

Reviewer #3:This manuscript has multiple conceptual and technical weaknesses:The interaction between 53BP1 and TopBP1 and its role in the G1 checkpoint has already been reported (Cescutti et al., 2010). Although this study provides some new details, the overall model is not significantly advanced.

It’s inaccurate to say that the whole story of the function of 53BP1 and TOPBP1 was already told in the work from the Hazalonetis laboratory (Cescutti et al., 2010). In that paper they showed that 53BP1 and TOPBP1 somehow collaborate and interact in G1 cells, and if very large chunks of TOPBP1 are deleted, the G1 checkpoint is damaged – it was that observation that originally piqued our interest in the interaction, and as you can imagine we have read it many times in great detail. What they showed was that the two proteins co-localised at DNA damage, probably interacted (although they did not show how), and that this co-localization/interaction was important. However, the downstream ‘work-up’ of that initial observation – now more than 10 years old – was limited. They did not succeed in localising the interaction at all on 53BP1 and only mapped it on TOPBP1 to the extent of seeing a checkpoint defect if BRCT45 was deleted. Whether the interaction was phosphorylation-dependent was not addressed, nor which arm of the main DNA damage signalling pathways (ATM-CHK2 or ATR-CHK1) that drive checkpoint activation was affected when the interaction was impaired in their gross deletions of TOPBP1. This is a key mechanistic detail without which the observation of a functional collaboration remains phenomenological and cannot be rationalised within the general understanding of the DNA damage response.

Starting from the baseline observations of Cescutti et al., in this manuscript we have:

a) precisely identified two key residues on 53BP1 (a 213kDa protein with more than 200 documented phosphorylation sites) that are required for the interaction with TOPBP1, and shown that this requires their DNA damage-dependent phosphorylation – not in Cescutti et al;

b) shown that *both* the BRCT012 *and* BRCT45 modules of TOPBP1 are required for the interaction by surgical point mutations – Cescutti et al. only identified BRCT45 and then only by gross domain deletion, where any observed loss of function could have been due to major effects on overall protein stability;

c) shown that the TOPBP1-53BP1 interaction is required to drive the ATR-CHK1 arm of the DDR but has no effect on ATM – not addressed in Cescutti et al;

d) shown for the first time that two phosphorylation-dependent ligand proteins of TOPBP1 with compatible BRCT-domain selectivity can brought into very close proximity by their mutual interaction with TOPBP1 at sites of DNA damage.

These are substantial and significant advances for the particular interaction of TOPBP1 and 53BP1, while d) opens up a whole new concept of the combinatorial complexity of TOPBP1-scaffolded multi-protein complexes that hasn’t so far been addressed anywhere else to our knowledge.

Furthermore, TopBP1 also interacts with MDC1 in a phosphorylation-mediated manner, which was also shown to regulate ATR activation (Wang et al., 2011 JCB). The structural details of the TopBP1-MDC1 interaction has been analyzed and reported (Leung et al., 2013). Given that the localization of 53BP1 to DNA damage sites is dependent on MDC1, the function of 53BP1 in ATR regulation must be regulated by MDC1. Overall, the results of this study only add new details to an existing model but do not advance the model significantly.

The reviewer suggests that our data and the original baseline observation of Cescutti et al., are irrelevant, as there is an alternative model (Wang et al., 2011 and Leung et al., 2013), whereby TOPBP1 recruitment and ATR activation (and by implication the G1 checkpoint) is actually dependent on MDC1. The reviewer suggests that this is a major conceptual weakness in our work.

In an earlier paper, from which this present manuscript partly stems, (Day et al., 2018), we showed clearly that at least in the fission yeast system, the model for MDC1 (Mdb1 in yeast) interaction with TOPBP1 (Rad4 in yeast) claimed in the Wang et al. and Leung et al. papers was not correct, and that the SDT motif suggested as recruiting TOPBP1/Rad4 to MDC1/Mdb1 actually interacts with the FHA domain of Nbs1.

We have subsequently gone on to show that this model of MDC1-TOPBP1 biochemical and functional interaction is also incorrect in the mammalian system. We have shown (Leimbacher et al., 2019) that none of the SDT motifs of MDC1 (claimed in Wang et al., 2011 and in Leung et al., 2013) interact with TOPBP1, and that BRCT45 plays no role in MDC1-TOPBP1 interaction. Instead, using the same methodology as in our earlier *eLife* and Molecular Cell papers and in this present manuscript, we identify novel phosphorylation sites on MDC1 that are critical for interaction with TOPBP1 in vitro and in vivo. These sites actually interact with BRCT1 and BRCT2 of TOPBP1, rather than BRCT5 and are likely to be mutually exclusive with 53BP1 and RAD9 in binding to TOPBP1. We further show that direct interaction of MDC1 and TOPBP1 is not required for the G1/S or G2 DNA damage checkpoints, but is instead critical for the response to DNA damage occurring in mitosis. The structural work cited by this reviewer on which part of the previous model for MDC1 interaction with TOPBP1 is based, is unreliable, as it is very likely that the MDC1-SDT interaction with TOPBP1-BRCT45 reported in Leung et al. is a crystallisation artefact. Indeed, to their credit, the authors of that work more or less concede this in a later paper (Sun et al., 2017) where they observe authentic binding of a BLM-derived phosphopeptide to BRCT45, which makes interactions almost identical to those we report here for the 53BP1-pS366 peptide. Thus, the notion of an alternative MDC1-dependent ATR activation that the reviewer presents as making our work conceptually weak, is actually incorrect.

This manuscript also suffers from several significant technical problems. For example, many experiments in this study used Cyclin A- as a marker for G1 cells. Cyclin A is largely a soluble nuclear protein. However, this study apparently used detergent-extracted cells for Cyclin A staining (Figure 4A). I am not sure if they have identified G1 cells correctly.

Cyclin A is widely used as a cell cycle stage marker due to its progressive up regulation once S-phase is established, maximal levels in G2, and rapid degradation at the onset of mitosis. Consequently, nuclei with diffuse chromatin staining and low levels of cyclin A immunoreactivity, can be reasonably assigned to the G1 phase of the cell cycle. This is an absolutely standard methodology we, colleagues in our institute, and many other labs in the genome stability and DNA repair field worldwide, use routinely.

The reviewer is concerned that our identification of G1 cells by this method is not reliable because our study ‘apparently used detergent-extracted cells for cyclin A staining (Figure 4A.)’. We did not use detergent-extracted cells: all the protocols used for nuclear protein immunofluorescence visualisation in this study are standard in the field and detailed in the Materials and methods. As is standard in the field, cells were first fixed with -20ºC methanol, which precipitates the proteins in situ, and then gently permeabilised for immunofluorescence. Methanol fixation is used for the explicit purpose of preventing any leaching of proteins that might occur due to permeabilization, and works extremely well, as the hundreds if not thousands of papers that use this technique for immunofluorescence microscopy would attest.

In Figure 4A we only showed cells without evident cyclin A staining as these are the G1 cells we are interested in analysing. However, as we are using asynchronous cultures, we of course have cells at different cell cycle stages on our slides, and accordingly see different levels of cyclin A immunofluorescence when our cells are fixed and prepared by this standard method. An example is included as Author response image 2.

**Author response image 2. respfig2:** 

There is abundant cyclin A signal in some cells which are present on the same slide as some cells which have undetectably low cyclin A signal. As all the cells on the slide experienced precisely the same fixation and permeabilisation protocol, which does not involve detergent extraction, it is clear that our preparation does not wash-out the cyclin A from the nucleus; if it did, none of the cells would show cyclin A.

The G1 checkpoint assay used in this study was not properly done. I don't see any BrdU+ EdU- cells in the si53BP1+IR sample in Figure 4C. I am not sure how the authors can conclude that 53BP1 knockdown abolished the G1 checkpoint. In Figure 4D, only BrdU labeling but not EdU/BrdU sequential labeling was done. This is not the proper G1 checkpoint assay. How can the authors be sure that these are G1 cells moving into S phase?

The G1 checkpoint assay works with asynchronous cell cultures, and uses a clever double pulse labelling protocol to identify cells that are already in S phase and those that are in G1 with the potential to enter S-phase. Firstly, cells are pulsed with EdU, which becomes incorporated into the DNA in any cells that are actively replicating – i.e. in S-phase – and can be recognised by an antibody. The cells are then subjected to DNA damage – in our case γ-irradiation – and treated with a second labelled nucleotide BrdU (which has its own distinctive antibody) and a mitotic blocker – we use nocodazole – that stops cells that are have already entered mitosis from dividing and going round again. This widely used protocol is fully documented in the Materials and methods.

Irradiated cells that are already in S-phase (and therefore EdU labelled) continue to synthesise DNA and also incorporate BrdU. Cells that were in G1 and therefore didn’t incorporate EdU prior to damage, stall at the G1 checkpoint so long as this is intact, and not enter S-phase and not therefore incorporate BrdU either. In normal circumstances therefore, you expect to see relatively few EdU-/BrdU+ cells – this is what we show in Figure 4C with U2OS cells and in Figure 4—figure supplement 1for RPE1 cells. If however, the cells are impaired in factors that contribute to the G1 checkpoint, such as the siRNA knockdown of 53BP1, EdU- G1 cells are not prevented from entering S-phase and start to incorporate BrdU. Thus, the presence of EdU-/BrdU+ cells following DNA damage, indicates a defect in the G1 checkpoint – and this is exactly how we have used that here.

To make the BrdU+/EdU- cells clear,we show individual channels as well as an EdU/BrdU merge in Author response image 3 with some EdU-/BrdU+ cells indicated by red arrows. In the revised manuscript we have amended this figure to explicitly show the EdU channel in top and bottom panels, included arrows to indicate the EdU-/BrdU+ cells of interest, and have included the separated channels figure shown in Author response image 3, in the supplementary figures.

**Author response image 3. respfig3:** 

The reviewer also raises a concern about Figure 4D relating to our use of this assay to determine the effect of mutations in the 53BP1 phosphorylation sites on G1 checkpoint function. The same protocol was used for all of these G1 checkpoint assays and in all cases involves sequential labelling with EdU and BrdU – this is presented in detail in the Materials and methods. The reason that no EdU signal is shown, is because the absence of EdU signal is what we use to select those cells that are not in S-phase and are therefore of interest if they show a BrdU signal. This is the essence of the assay. The EdU channel was of course recorded for all of the mutants analysed as this is the only way to identify those cells that are EdU-, but we didn’t show it, because in these zoomed-in images of EdU-/BrdU+ G1 cells there was, by definition, nothing on it.

However, in Figure 4—figure supplement 2B we show the individual channels underlying, including the EdU channel. There is an EdU+/BrdU+ (and therefore S phase cell) visible on the siRNA+WT add-back panel, but there is no EdU signal in the other panels which are focussed on EdU- cells. In the revised manuscript this figure shows the EdU signal.

Additionally, in Figure 4—figure supplement 1, we repeat the entire exercise in RPE1 cells using the same double labelling G1 checkpoint assay.

In Figure 4—figure supplement 1B – the equivalent of Figure 4C in the main figures – distinct EdU+ and EdU- cells can be seen in the top panels, and distinct EdU-/BrdU+ cells (indicated with red arrows) in the bottom panels. In Figure 4—figure supplement 1D – the equivalent of Figure 4D in the main figures – we show the EdU channel as well as the BrdU channel, and some small signal for this is visible in a couple of the frames. Of course, if you focus in on the cells of interest, those that are EdU-/BrdU+, by definition there is no EdU signal visible – but we still did the EdU pulse. As with the U2OS data in all cases we used the standard sequential double labelling protocol set out in detail in the Materials and methods.

In Figure 3C, why did anti-pT670 antibody detect nuclear foci that did not colocalize with eYFP-53BP1?

Many antibodies, both commercial and lab-commissioned as here, show some off-target affinity – this is not unusual. The off-target interaction does not colocalise with 53BP1 and therefore has no bearing on the on-target phospho-specific interaction with pT670 which we fully validate. This antibody is not used for any of the co-localisation studies where an off-target interaction could be a problem, but only to verify that Thr670 is phosphorylated following DNA damage in vivo.

In Figure 5A, pATR formation is not defective in cells expressing the TopBP1 T670A mutant. pATR just did not colocalize with eYFP-53BP1. Where is pATR in these cells? Would this result argue against the requirement of TopBP1 phosphorylation at T640 for ATR activation?

This is not correct: the T670A mutation was introduced into 53BP1 as is clearly stated in the manuscript. We do not understand the reviewer’s reference to phosphorylation of TOPBP1 at Thr640 as being required for ATR activation. We are not aware of any such phosphorylation being suggested to be important for ATR activation. In human TOPBP1 residue 640 maps to the N-terminus of BRCT domain 5 and is well away from the genetically mapped ATR-activating domain which occurs between BRCT6 and BRCT7,8 and is in any event a methionine rather than a threonine.

We do not know the location of pATR molecules that do not co-localize with 53BP1, but it is reasonable to suppose that some of them remain co-located with TOPBP1, which no longer colocalises with 53BP1 when 53BP1 is mutated on T670 and/or S366. They may also be interacting with ETAA1 – a recently described alternative ATR activating molecule that functions independently of the RAD-RFC – 9-1-1 – TOPBP1 system. In budding yeast there is also evidence for direct ATR activation by RAD-RFC – 9-1-1 independent of TOPBP1, but no clear data supporting such an alternative has emerged for the metazoan system.

In Figure 5C, it is surprising that some many Chk1 foci were detected. Chk1 is mobile protein after DNA damage.

It isn’t clear whether the reviewer is questioning whether ‘some’ CHK1 foci were detected, or whether ‘so many’. The reviewer’s statement about CHK1 mobility likely stems from a paper from the Bartek laboratory where they looked at fluorescently-tagged CHK1 and CHK2 recruitment to UV laser stripe damage in live cells, but only over minutes after the DNA damage – laser stripe studies cannot address longer time scales as the live cells move around too much. Furthermore, the levels of slowly-repairing complex double-strand breaks generated by laser damage are rather low. In longer time scale observations (we use 4 hours) following ionising radiation, which generates high-levels of slowly-repaired chemically complex DSBs, persistent CHK1 and ATR foci are routinely observed in both U2OS and RPE1 cells. We are certainly not the only people to see this and cited three references to previous observations of CHK1 foci in the manuscript – so it is not surprising if you look at the more recent literature.

In Figure 6F, there are no controls to check the effects of siTopBP1 on 53BP1 and RAD9 foci. Both 53BP1 and RAD9 are known to localize to DNA damage sites through TopBP1-independent mechanisms. How does TopBP1 affects the 53BP1-RAD9 colocalization?

We have of course performed controls but we did not show these in the original submission. TOPBP1 knockdown has a negligible effect on 53BP1 foci numbers observed after irradiation in G1 cells. While TOPBP1 knockdown does decrease the total number of RAD9 foci, plenty of RAD9 foci remain, however consistent with the PLA results these show a decreased degree of coincidence with 53BP1. These controls have been added as a supplementary figure.

We are surprised by the question in the last part of this comment as the effect of TOPBP1 on colocalization of 53BP1-RAD9 was clearly shown in the Proximity Ligation Assay data presented in Figure 6F in the manuscript. Here siRNA knockdown of TOPBP1 can be seen to cause a very significant decrease in the frequency with which 53BP1 and RAD9 molecules colocalise with each other. TOPBP1 therefore appears to promote the co-localisation of these molecules, and the 40nm cut-off of the PLA system suggests that this is most likely due to simultaneous interaction with the same TOPBP1 molecule.

The model in Figure 7 is not supported by data.

Each process and interaction shown schematically in Figure 7 and expanded upon in the accompanying figure legend, is very well supported in the published literature, or as in the case of the phosphorylation-dependent interaction of TOPBP1 and 53BP1, is demonstrated in this manuscript. The only exception to this is the identity of the kinase (or kinases) that is (are) responsible for the phosphorylation of Ser366 and Thr670 on 53BP1 following DNA damage, which we have not yet definitively determined. As is entirely reasonable for the Discussion section of a paper, the figure does involve one small speculation for which no data is yet available, in that we show the two complementary interactors with TOPBP1, the 9-1-1 clamp, and 53BP1 associated with the opposite sides of a DSB, rather than the same side. However, in our defence we do clearly state in the figure legend that we don’t yet know which is the case.

What is the evidence that 53BP1 recruits TopBP1? If 53BP1 recruits TopBP1 in G1 cells, why are 53BP1 and RAD9 colocalized?

Our model does not claim that 53BP1 recruits TOPBP1 as such, but rather as we show it in Figure 7TOPBP1 is simultaneously ‘recruited’ by 53BP1 and RAD9, both of which it interacts with, scaffolding their colocalisation.

What is the evidence that 53BP1 functions independently of RAD9 to regulate ATR? I don't understand the logic of the model proposed.

Neither 53BP1 nor RAD9 directly regulate ATR, rather ATR activation depends on interaction with TOPBP1, which in the best understood S/G2 model is tethered in the vicinity of DNA damage (in the form of RPA-coated ssDNA generated by MRN) by the phosphorylated tail of RAD9 loaded along with RAD1 and HUS1 at a 5’-recessed dsDNA-ssDNA junction. In the G1 model we explore here we show that TOPBP1 also interacts with 53BP1, which could in principle provide an alternative tether. However, as we go on to show, this interaction – which is critical in G1 – likely occurs alongside interaction with RAD9. Although it was previously thought that no resection would occur in G1 DSBs, which are overwhelmingly repaired by NHEJ, recent work from Lobrich, Jeggo and others clearly shows that all DSBs undergo some degree of resection to generate short segments of RPA-coated ssDNA. Our model therefore provides a framework for understanding how TOPBP1 helps integrate the presence of 53BP1 – which is strongly recruited downstream of DSB recognition by MRN and ATM signalling – with the presence of 9-1-1 recruited to the dsDNA-ssDNA junction generated by resection – to activate the ATR-CHK1 arm of the DNA damage response.